# Single Model Uncertainty Estimation via Stochastic Data Centering

**Jayaraman J. Thiagarajan***
Lawrence Livermore National Laboratory
jjayaram@llnl.gov

**Rushil Anirudh***
Lawrence Livermore National Laboratory
anirudh1@llnl.gov

**Vivek Narayanaswamy**
Arizona State University
vnaray29@asu.edu

**Peer-Timo Bremer**
Lawrence Livermore National Laboratory
bremer5@llnl.gov

## Abstract

We are interested in estimating the uncertainties of deep neural networks, which play an important role in many scientific and engineering problems. In this paper, we present a striking new finding that an ensemble of neural networks with the same weight initialization, trained on datasets that are shifted by a constant bias gives rise to slightly inconsistent trained models, where the differences in predictions are a strong indicator of epistemic uncertainties. Using the neural tangent kernel (NTK), we demonstrate that this phenomena occurs in part because the NTK is not shift-invariant. Since this is achieved via a trivial input transformation, we show that this behavior can therefore be approximated by training a single neural network – using a technique that we call $\Delta-$UQ – that estimates uncertainty around prediction by marginalizing out the effect of the biases during inference. We show that $\Delta-$UQ 's uncertainty estimates are superior to many of the current methods on a variety of benchmarks– outlier rejection, calibration under distribution shift, and sequential design optimization of black box functions. Code for $\Delta-$UQ can be accessed at github.com/LLNL/DeltaUQ

## 1 Introduction

Accurately estimating uncertainties in a deep neural network (DNN) is an active area of research due to its implications in a wide range of scientific and engineering problems. Broadly, there are two kinds of uncertainties that are often considered – (a) *aleatoric*: uncertainties in the data generating process that are typically irreducible, and (b) *epistemic*: uncertainties of the model that can be reduced by observing more data, which is the focus of our work. Some of the popular techniques for the latter include Bayesian methods [1–4] that use a prior on the network weights, Monte Carlo approximations such as MC Dropout [5] that approximate sampling from the posterior, and empirical methods such as Deep Ensembles (DEns) [6]. In particular, DEns trains an ensemble of neural networks with different initializations, such that the uncertainty estimate on a test sample is given by the inconsistency between predictions from the member models. In practice, DEns has been found to often outperform other related methods [7, 6, 8], but it comes at the cost of training several DNNs to obtain reliable uncertainties (typically $10-20$), which is a severe computational bottleneck when it comes to modern DNNs especially on large scale datasets. In light of this, there is increased interest in developing single model estimators that can still produce high quality uncertainties. There has been some promising work in this direction in the recent past, specific to deep classifiers [8] or regression models [9], and has been shown to perform comparably to DEns in some use-cases.

---

*equal contribution

36th Conference on Neural Information Processing Systems (NeurIPS 2022).

In this work, we first begin by exploring an alternate approach for constructing deep ensembles – instead of using multiple randomized weight initializations, we propose to shift the training domain using random biases, such that every model in the ensemble is trained with data that has been shifted by a different *constant* bias c. Formally, for a labeled dataset $\{(x, y)\}$, the $k^{\text{th}}$ model is trained to fit $\{(x - c_k, y)\}$, where $c_k$ (referred as an anchor) is of the same size as x. Though this manipulation appears trivial, the kernel induced by deep networks is not inherently shift invariant [10, 11], thus implying each DNN learns a slightly different model due to the bias. This leads to one of our key observations:

> **Anchor Ensemble:** *When an ensemble of DNNs, with the same fixed initialization, is trained on a dataset shifted by random constant biases, the variation across the ensemble's predictions is a strong indicator of model uncertainty.* $\boxed{1}$

Based on analysis with the neural tangent kernel (NTK) [10], we show that when the anchor c is made a random variable, the corresponding kernel is stochastic such that for each c, the model converges to a slightly different NTK. Consequently, this anchoring-based ensembling provides a different approach to DEns for sampling solutions from the hypothesis space. However, interestingly, since the anchor ensembling emerges primarily from the act of a trivial data transformation, it lends itself to be approximated easily using a single DNN:

> $\Delta-$**UQ** : *For a randomly chosen anchor* c, $\boxed{1}$ *can be approximated using a single DNN trained on the dataset transformed as* $\{x, y\} \rightarrow \{[c, x - c], y\}$. $\boxed{2}$

During inference, we obtain multiple predictions for a given sample by varying the choice of the anchor, such that the standard deviation of the predictions is our estimate for uncertainty. Without affecting the performance of the model, we find that $\Delta-$UQ produces meaningful epistemic uncertainties that we validate in a variety of applications: outlier rejection, calibration under distribution shifts on ImageNet, and sequential optimization of a large suite of black-box functions. We observe that $\Delta-$UQ consistently outperforms existing uncertainty estimates, while also being efficient to train as a single model estimator. With just a few simple changes, one can easily modify the training of any existing DNN to support $\Delta-$UQ estimation.

## 2   Background and Related Work

Denote training data as $\mathcal{D} = \{(x_i, y_i)\}_{i=1}^n$, where $x_i \in \mathcal{X}$ and $y_i \in \mathcal{Y}$, to train a neural network $f(\boldsymbol{\theta}) \in \mathcal{H}$ with randomly initialized weights $\boldsymbol{\theta}_0$, such that a loss function $\mathcal{L}$ is minimized, *i.e.*, $\arg\min_{\boldsymbol{\theta}} \mathcal{L}(f(x; \boldsymbol{\theta}), y)$. Here, $\mathcal{H}$ denotes the hypothesis space of potential solutions for fitting the observed data. Given a prior distribution on the weights $p(\boldsymbol{\theta})$, we can define the posterior over $\boldsymbol{\theta}$ as $p(\boldsymbol{\theta}|\mathcal{D})$ and subsequently, infer the posterior predictive distribution for a test sample $(x_t, y_t)$. This can be used to quantify the uncertainty around the prediction as $p(y_t|x_t, \mathcal{D}) = \int_{\boldsymbol{\theta}} p(y_t|x_t, \boldsymbol{\theta})p(\boldsymbol{\theta}|\mathcal{D})d\boldsymbol{\theta}$.

A challenge for neural networks, however, is that the posterior $p(\boldsymbol{\theta}|\mathcal{D})$ is computationally intractable. This has motivated the use of *Bayesian Neural Networks* (BNNs) [3] using different approximations to the posterior including Monte-Carlo Dropout [5], variational inference [12, 4], and sampling methods such as Markov chain Monte Carlo [3, 13]. In parallel, it has been empirically shown that Deep Ensembles (DEns) [6] often tend to outperform Bayesian methods in terms of model calibration performance, even under challenging distribution shifts [7]. The success of DEns has been attributed to its ability to sample different functional modes [14] from the hypothesis space, and thus approximate the posterior predictive distribution [1]. However, a critical limitation of DEns is the need to train multiple models (typically $10 - 20$) in order to obtain well calibrated uncertainties, which can be impractical for complex model architectures that have become commonplace today.

Characterizing the behavior of deep uncertainty estimators has mostly been done using empirical evaluation based on model calibration or out-of-distribution detection, but the recent advances in the neural tangent kernel (NTK) theory [10, 15–17] provide a convenient framework for more rigorous analysis. The basic idea of NTK is that, when the width of a neural network tends to infinity and the learning rate of stochastic gradient descent (SGD) tends to zero, the function $f(x; \boldsymbol{\theta})$ converges to a solution obtained by kernel regression using the NTK defined as $\mathbf{K}_{x_i x_j} = \mathbb{E}_{\boldsymbol{\theta}} \left\langle \frac{\partial f(x_i, \boldsymbol{\theta})}{\partial \boldsymbol{\theta}}, \frac{\partial f(x_j, \boldsymbol{\theta})}{\partial \boldsymbol{\theta}} \right\rangle$. When the samples $x_i, x_j \in \mathcal{S}^{d-1}$, i.e., points on the hypersphere and have unit norm, the NTK for a

simple 2 layer ReLU MLP can be simplified as a dot product kernel [15–17]:

$$\mathbf{K}_{\mathbf{x}_i \mathbf{x}_j} = h_{\mathrm{NTK}}(\mathbf{x}_i^\top \mathbf{x}_j) = \frac{1}{2\pi} \mathbf{x}_i^\top \mathbf{x}_j (\pi - \cos^{-1}(\mathbf{x}_i^\top \mathbf{x}_j)) \tag{1}$$

The NTK framework and its extensions that enable a posterior interpretation in the infinite limit have been used to study deep ensembles [2]. Previous work [18–20] has also shown the existence of a distinct yet related kernel, referred to as the neural network Gaussian process (NNGP) kernel, where the initialization tends to a GP in infinite width limit.

# 3 Uncertainty estimation with $\Delta-$UQ

As discussed above, most existing frameworks sample the hypothesis space either through different random initializations of ($\boldsymbol{\theta_0}$), perturbing the weight space after training, or using Bayesian methods by placing a prior on $p(\boldsymbol{\theta})$. Here, we propose to use a new strategy that involves injecting multiple trivial biases into the training data and analyze the resulting models using the NTK framework.

## 3.1 Anchor Ensembles: ensembling by injecting trivial biases

Let us examine the scenario where we shift an entire dataset (both train and validation) using a constant bias, $c$, to obtain a new dataset $\mathcal{D}_c$ using which we train the model $f_c$. Since we always choose $c$ from the training distribution at random, this has the effect of zero-centering the dataset around different training points. Let $\{f_{c_1}, f_{c_2}, \ldots, f_{c_k}\}$ denote the set of models trained using different $c$'s, then our goal is to characterize the relationship between them as a function of $c$. If the NTK induced by $f$ is shift-invariant (for e.g., when Fourier features [11] are used), the shifts will make no difference, resulting in identical models $f_{c_1} = \cdots = f_{c_k}$. However, since NTKs for models like MLPs and CNNs are not inherently shift-invariant [17][2], we find that the models lead to an effective deep ensemble, wherein the variation across the predictions is a strong indicator of epistemic uncertainties.

**Effect of shifted training on NTK:** We are interested in understanding how (1) changes when the *entire training domain* is shifted by $c$ – i.e., $h_{\mathrm{NTK}}((\mathbf{x}_i - c)^\top(\mathbf{x}_j - c))$. Without loss of generality, for the sake of notational convenience, we assume $\mathbf{x}_i - c$ and $\mathbf{x}_j - c$ are also made unit norm. To simplify the expansion, we use a Taylor series expansion for the $\cos^{-1}$ function: $\cos^{-1}(u - c) \approx \cos^{-1}(u) + \frac{c}{\sqrt{1-(u-c)^2}}$.

Expanding $(\mathbf{x}_i - c)^\top(\mathbf{x}_j - c)$ as $\mathbf{x}_i^\top \mathbf{x}_j - c^\top(\mathbf{x}_i + \mathbf{x}_j - c)$ and letting $v = (\mathbf{x}_i + \mathbf{x}_j - c)$, we obtain the expression for $h_{\mathrm{NTK}}$ under a shifted domain as follows:

$$\mathbf{K}_{(\mathbf{x}_i - c)(\mathbf{x}_j - c)} = \frac{1}{2\pi}(\mathbf{x}_i^\top \mathbf{x}_j - c^\top v)(\pi - \cos^{-1}(\mathbf{x}_i^\top \mathbf{x}_j - c^\top v))$$

$$\approx \frac{1}{2\pi}\mathbf{x}_i^\top \mathbf{x}_j (\pi - \cos^{-1}(\mathbf{x}_i^\top \mathbf{x}_j)) - \frac{1}{2\pi}c^\top v(\pi - \cos^{-1}(\mathbf{x}_i^\top \mathbf{x}_j)) - \frac{c(\mathbf{x}_i^\top \mathbf{x}_j - c^\top v)}{2\pi\sqrt{1 - (\mathbf{x}_i^\top \mathbf{x}_j - c^\top v)^2}}$$

$$= \mathbf{K}_{\mathbf{x}_i \mathbf{x}_j} - \Gamma_{\mathbf{x}_i, \mathbf{x}_j, c}, \tag{2}$$

where we combine all terms dependent on $c$ into $\Gamma_{\mathbf{x}_i, \mathbf{x}_j, c}$, which also behaves as a dot product kernel. From (2), we note that a trivial shift in the domain results in a non-trivial shift in the NTK function itself. In other words, (2) outlines the *effective* NTK as a function of $c$. We also note that $\Gamma$ does not affect the spectral properties of the original NTK, as we observe in Figure 1.

Let us now consider the prediction on a test sample $\mathbf{x}_t$ in the limit as the inner layer widths grow to infinity. It has been shown that (c.f. [17, 16]):

$$f_\infty(\mathbf{x}_t) = f_0(\mathbf{x}_t) - \mathbf{K}_{\mathbf{x}_t \mathbf{X}} \mathbf{K}_{\mathbf{X}\mathbf{X}}^{-1}(f_0(\mathbf{X}) - \mathbf{Y}), \tag{3}$$

---

[2]By shift-invariance, we refer to the *domain* shift induced because of the random bias, as opposed to *spatial* shift-invariance that is more commonly associated with convolutional operators.

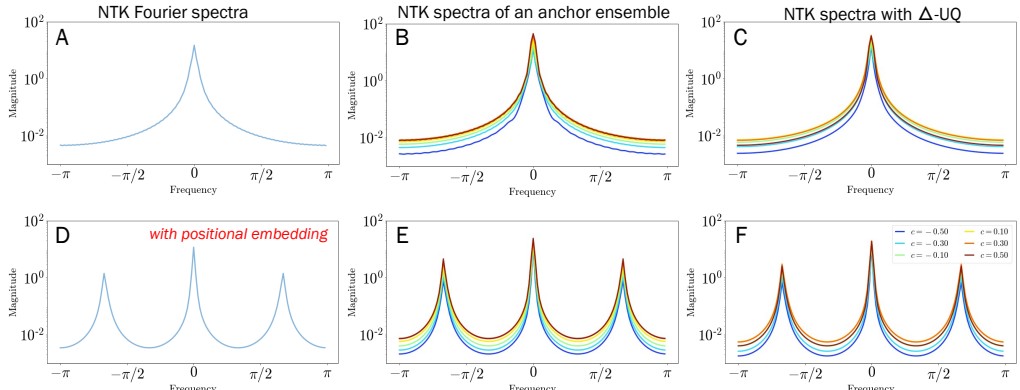

Figure 1: Fourier spectrum of an NTK for an MLP model (A,D); spectra of an anchor ensemble (B, E); and NTK spectra using $\Delta-$UQ (C, F). Bottom row shows NTK spectra when inputs are passed through a sinusoidal PE. We make two key observations – a) trivial shifts in the input domain cause the *effective* NTK to be distinct as a function of the shift $c$, as seen in eqn. (2); and b) $\Delta-$UQ achieves a similar effect but with a single model.

where $\mathbf{X}$ is the matrix of all training data samples. As before, we consider the case where the domain is shifted by $c$. Using (3):

$$
\begin{aligned}
f_\infty(\mathrm{x}_t - \mathrm{c}) &= f_0(\mathrm{x}_t - \mathrm{c}) - \mathbf{K}_{(\mathrm{x}_t-\mathrm{c})(\mathbf{X}-\mathrm{c})}\mathbf{K}^{-1}_{(\mathbf{X}-\mathrm{c})(\mathbf{X}-\mathrm{c})}(f_0(\mathbf{X} - \mathrm{c}) - \mathbf{Y}) \\
&\approx f_0(\mathrm{x}_t - \mathrm{c}) - (\mathbf{K}_{\mathrm{x}_t\mathbf{X}} - \Gamma_{\mathrm{x}_t,\mathbf{X},\mathrm{c}})(\mathbf{K}_{\mathbf{XX}} - \Gamma_{\mathbf{X},\mathbf{X},\mathrm{c}})^{-1}(f_0(\mathbf{X} - \mathrm{c}) - \mathbf{Y}) \\
&= f_0(\mathrm{x}_t - \mathrm{c}) - (\mathbf{K}_{\mathrm{x}_t\mathbf{X}} - \Gamma_{\mathrm{x}_t,\mathbf{X},\mathrm{c}})\left(\mathbf{K}^{-1}_{\mathbf{XX}} + \sum_{m=1}^\infty (\mathbf{K}^{-1}_{\mathbf{XX}}\Gamma_{\mathbf{X},\mathbf{X},\mathrm{c}})^m \mathbf{K}^{-1}_{\mathbf{XX}}\right)(f_0(\mathbf{X} - \mathrm{c}) - \mathbf{Y})
\end{aligned}
$$
(4)

$$
\approx \underbrace{f_0(\mathrm{x}_t) - \mathbf{K}_{\mathrm{x}_t\mathbf{X}}\mathbf{K}^{-1}_{X\mathbf{X}}(f_0(\mathbf{X}) - \mathbf{Y})}_{\text{deterministic for fixed } \boldsymbol{\theta}_0} - \underbrace{g(\mathrm{c}, \mathrm{x}_t, \mathbf{X}, \mathbf{Y})}_{\text{random due to c}}
$$
(5)

In (4), we utilize Woodbury's Inverse Identity [21] to expand the inverse of a sum of matrices. Next, in (5), we combine all the terms dependent on $c$ into a function $g$. Note, we also expand $f_0(\mathrm{x} - \mathrm{c})$ using the Taylor series approximation to represent it as a sum of $f_0(\mathrm{x})$ and other terms (details of this derivation can be found in the appendix).

For a given initialization $\boldsymbol{\theta}_0$, the first term is deterministic, and exactly the same as (3), since all other the terms are fixed. However, we see that the second term can vary based on the choice of $c$. Existing ensembling approaches [6] rely on the randomness of the initialization $\boldsymbol{\theta}_0$ to pick diverse solutions from the hypothesis space [2]. In contrast, equations (2) and (5) suggest that even for a fixed $\boldsymbol{\theta}_0$, it is possible to make the NTK stochastic (in $c$) by shifting the entire input domain. To better understand how the NTK actually changes from these expressions, we study the spectral properties of a simple MLP network empirically, following the analysis in [11].

**Spectral properties of shifted NTKs.** We compute the Fourier spectra using the same MLP on several shifted domains in Figure 1(B). The original spectrum for the MLP without any shift in the training domain is shown for comparison in 1(A). As indicated by (2), we see that each individual shift leads to a different NTK (as indicated by the spectra), either by flattening it or making it narrower in the frequency domain. The same behavior persists even when we construct positional embeddings (PE), based on sinusoidal functions, prior to building the MLP model (1(D-E)). This leads to one of our main findings stated in $\boxed{1}$, and illustrated in Figure 3.

## 3.2   $\Delta-$**UQ : rolling anchor ensembles into a single model**

Since different models in an anchoring-based ensemble are trained with the same initialization, we present a new technique to approximate the epistemic uncertainties using a single neural network. More specifically, we perform a simple coordinate transformation by lifting the domain to a higher dimension as $\mathcal{E} : \mathrm{x} \to \{\mathrm{c}, \mathrm{x} - \mathrm{c}\}$, we refer to the residual by $\Delta = \mathrm{x} - \mathrm{c}$. This transformation

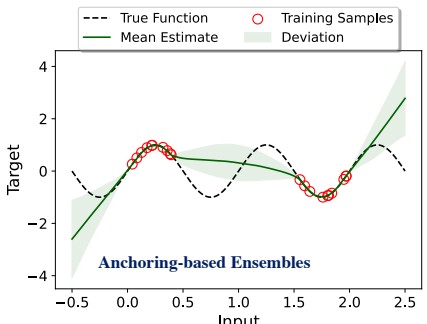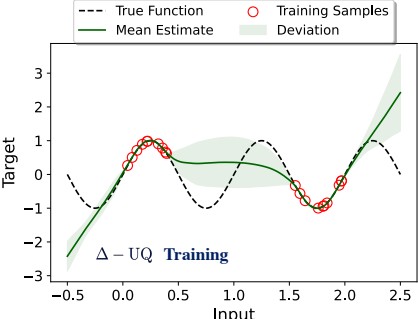

Figure 3: Comparing anchor ensembles and $\Delta-$UQ in function fitting with an MLP. As expected, we see that the disagreement between models in an anchor ensemble correlate strongly with the epistemic uncertainty, and that $\Delta-$UQ , with a single model, matches this behavior very closely.

allows the use of multiple representations (w.r.t. different anchors) for the same input sample x, *i.e.*, $f_\Delta(\{c_1, x - c_1\}) = f_\Delta(\{c_2, x - c_2\}) = \cdots = f_\Delta(\{c_k, x - c_k\})$, where $f_\Delta$ refers to the $\Delta-$UQ model that takes the tuple $(\{c_k, x - c_k\})$ and predicts the target $y$.

It is easy to see that $[c, x_i - c]^\top [c, x_j - c] = x_i^\top x_j - c^\top(x_i + x_j - 2c)$. That is, the dot product of the transformed inputs takes the same form as before (except for a scaling factor). Therefore, the expressions for the equivalent NTK for different anchor shifts, seen in (2), and the corresponding prediction on a test sample seen in (5) remain the same for $\Delta-$UQ , by setting $v = x_i + x_j - 2c$. A similar form is also obtained in the more general case when different anchors are used for different samples (as shown in the appendix), which leads to our main claim stated in $\boxed{2}$, that the $\Delta-$UQ model achieves similar perturbations of the NTK as an anchor ensemble, based on the choice of c.

**Training.** During training, for every input $x_i$ we choose an anchor as a random sample from the training dataset. Subsequently, we obtain the coordinate transformation $\{[c, x_i - c], y_i\}$, using which we train the model. With vector-valued data, this is implemented as a simple concatenation. In the case of image data, we append the channels to create a $6-$dimensional tensor (for a 3-channel RGB image). We show simple a PyTorch snippet for training $\Delta-$UQ in Figure 2. Other than increasing the number of parameters in the first layer of the network, $\Delta-$UQ does not incur additional computational overheads.

Over the course of training, in expectation, every training pair gets combined with a large number of anchors. Since the prediction on this training pair – regardless of anchor choice – must always be the same, this places an implicit consistency in the predictions that they must be similar no matter which anchor is chosen. This consistency trades-off with diversity of the kinds of functions that can be learned when compared with an anchor ensemble, where the models are trained independently.

Figure 2: Mini-batch training with $\Delta-$UQ .

```
for inputs, targets in trainloader:
    A = Shuffle(inputs) %% Anchors
    D = inputs-A %% Delta
    X_d = torch.cat([A, D],axis=1)
    y_d = model(X_d) %% prediction
    loss = criterion(y_d,targets)
```

This can be seen in the comparisons of the NTK spectra for $\Delta-$UQ with anchor ensembles in Figures 1(C) and (F). In practice, however, we find that the diversity from this single model is still sufficiently large, to estimate good quality uncertainties.

**Inference.** For a test sample $x_t$, we obtain the prediction from $\Delta-$UQ as the mean prediction across several randomly chosen anchors; and the standard deviation around these predictions is our estimate for the epistemic uncertainty. In other words, we marginalize out the effect of anchors to obtain the final prediction mean and uncertainty. Formally, the predictive distribution is given by $p(y_t|x_t) = \int_{c \in \mathbf{X}} p(y_t|x_t, c, \boldsymbol{\theta})p(c)dc$. In practice, for a trained $\Delta-$UQ model specified as $\boldsymbol{\theta}^*$, we compute the sample mean and uncertainty around it as:

$$\boldsymbol{\mu}(y_t|x_t) = \frac{1}{K}\sum_{k=1}^{K} f([c_k, x_t - c_k], \boldsymbol{\theta}^*); \quad \boldsymbol{\sigma}(y_t|x_t) = \sqrt{\frac{1}{K-1}\sum_{k=1}^{K}(f([c_k, x_t - c_k], \boldsymbol{\theta}^*) - \boldsymbol{\mu})^2},$$

$$(6)$$

**Discussion.** In Figure 3, we show a 1D regression example using 20 training examples along with the predicted mean and estimated uncertainties. As it can be seen, both the anchoring-based ensemble (left) and $\Delta-$UQ training show higher epistemic uncertainties around regions with no training samples. In the ensembles version, we train 20 different networks – while in $\Delta-$UQ we use just a single network, where the uncertainty is obtained using all 20 anchors during inference.

We always use only a single anchor for an input during every training iteration, but multiple anchors during inference time. In theory, multiple anchors could be used during training as well, where the loss is imposed on the mean (obtained with multiple anchors). However, we find that simply using a single random anchor in each iteration achieves a similar effect, as it enforces a consistency that the same training pair $(x, y)$ when combined with many different anchors over the course of training as $[c_1, x_i - c_1], [c_2, x_i - c_2], \ldots, [c_k, x_i - c_k]$ must all produce the same prediction, $y$.

$\Delta-$UQ relies on randomly drawn anchors for uncertainty estimation, which is similar to DEns [6], that relies on the diversity of the base learners in an ensemble. Arguably, sampling a random set of anchors from the training distribution is simpler than sampling from the posterior $p(\boldsymbol{\theta}|\mathcal{D})$. Furthermore, since every anchor realizes a slightly different function, using a small number of anchors $(10 - 20)$ during inference is typically sufficient to obtain high quality estimates as we show in our experiments. Finally, due to the nature of training with random anchors we also see that $\Delta-$UQ produces particularly effective uncertainty estimates when the training set size is small, and this proves to be very useful in applications such as sequential optimization.

## 4 Experiments

We validate our approach in this section using a variety of applications and benchmarks – (a) first, we consider the utility of epistemic uncertainties in object recognition problems where they have been successfully used for outlier rejection and calibrating models under distribution shifts. We show that $\Delta-$UQ can be very effective even with large-scale datasets like ImageNet [22]; (b) next, we consider the challenging problem of sequential design optimization of black-box functions, where the goal is to maximize a scalar function of interest with the fewest number of sample evaluations. Using a Bayesian optimization setup, we show that the uncertainties obtained using $\Delta-$UQ outperform many competitive methods across an extensive suite of black-box functions.

**a. Outlier rejection.** A popular application for epistemic uncertainties is in rejecting outliers since, by definition, they are in regions outside of the training distribution. As such, we expect the model to produce highly uncertain predictions for these images, which should help us design an effective OOD detector. To evaluate this hypothesis, we train a modified ResNet-50 [23] model on ImageNet that accepts 6 input channels (anchor, $\Delta$) as outlined earlier. We train the model using standard hyperparameter settings except, training it longer for 120 epochs – our top-1 accuracy (76.1) matches that of a standard ResNet-50. Specifically for images, we found that corrupting the anchors with common transforms like random crops, Gaussian blurs, and color jitter improves performance. That is, instead of $[c, x - c]$, we use $[\mathcal{T}(c), x - c]$ where $\mathcal{T}$ is a simple transform like Gaussian blur. The uncertainty for a test sample is given by the standard deviation of the logits obtained by varying the anchors. To obtain a scalar statistic, we compute the mean across all classes as the uncertainty score for that sample. We provide more details on the transformations used for corrupting the anchor, and the accuracies of our model across shifted variants in more detail in the appendix.

We show the results for outlier rejection in 4b, where we follow the protocol established in [25], that uses a Gaussian blur of intensity 5 from ImageNet-C [26] as the outlier set, and the clean ImageNet validation data as inliers. We use the estimated uncertainty obtained with 10 anchors as in (6) as our score for outlier rejection and report commonly used metrics such as AUROC, Detection Accuracy (DTACC), and AUPR-in/out. We note that, just the inconsistency of predictions obtained using $\Delta-$UQ outperforms many baselines including mean-field stochastic variational inference (SVI) [12, 4], SVI-AvUC [25], Monte Carlo dropout [5], and temp. scaling [24]. While this can be further improved by taking the mean prediction into account, similar to existing approaches for semantic novelty detection (with scores such as entropy, energy [27] etc.), our focus here is to evaluate the quality of uncertainty alone. In Figure 4a, we observe that $\Delta-$UQ 's uncertainty estimate changes smoothly as the outliers become farther away (more severe intensity) from the training distribution.

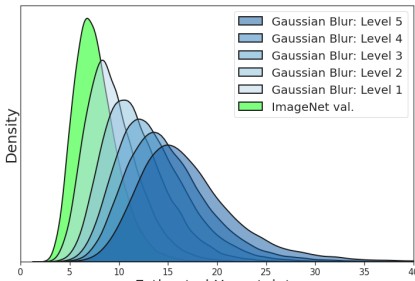

(a) Uncertainties change meaningfully as out-liers get more severe

| Method | AUROC↑ | DTACC↑ | AUPR-in/out↑ |
|---|---|---|---|
| ResNet-50 [23] | 93.36 | 86.08 | 92.82 / 93.71 |
| Temp-Scal [24] | 93.71 | 86.47 | 93.21 / 94.01 |
| Deep Ens [6] | 95.49 | 88.82 | 95.31 / 95.64 |
| MC Dropout [5] | 96.38 | 89.98 | 96.16 / 96.67 |
| SVI [4] | 96.40 | 90.03 | 95.97 / 96.83 |
| $\Delta-$UQ (ours) | **97.49** | **91.90** | **97.56 / 97.47** |

(b) Uncertainties from $\Delta-$UQ for outlier rejection

Figure 4: **Rejecting outliers with epistemic uncertainties:** We evaluate $\Delta-$UQ on the benchmark introduced by [25] where we use Gaussian Blur of level 5 intensity as the outliers from the ImageNet validation set. At inference, uncertainties are estimated as the sum of std. dev of predictions obtained with 10 anchors.

Table 1: **Calibration under distribution shift:** A ResNet-50 model that is tempered by uncertainties obtained from $\Delta-$UQ (see text) outperforms several competitive baselines averaged across 16 different corruptions of ImageNet-C at highest severity level 5.

| Metric | | Vanilla | Temp Scaling | DEns | MCD | LL-Dropout | SVI | SVI-AvUC | Ours |
|---|---|---|---|---|---|---|---|---|---|
| ECE↓ | lower quartile | 0.124 | 0.096 | 0.050 | 0.078 | 0.093 | 0.072 | 0.032 | 0.022 |
| | median | 0.174 | 0.139 | 0.090 | 0.134 | 0.145 | 0.114 | 0.045 | 0.038 |
| | mean | 0.194 | 0.160 | 0.088 | 0.153 | 0.161 | 0.119 | 0.054 | 0.044 |
| | upper quartile | 0.274 | 0.236 | 0.126 | 0.219 | 0.236 | 0.172 | 0.070 | 0.063 |
| NLL↓ | lower quartile | 4.635 | 4.53 | 4.035 | 4.699 | 4.563 | 4.322 | 4.164 | 4.014 |
| | median | 5.115 | 4.993 | 4.624 | 5.093 | 5.034 | 4.853 | 4.823 | 4.617 |
| | mean | 5.234 | 5.091 | 4.604 | 5.553 | 5.201 | 4.865 | 4.707 | 4.352 |
| | upper quartile | 6.292 | 6.165 | 5.893 | 6.522 | 6.342 | 6.034 | 5.778 | 4.987 |
| Brier↓ | lower quartile | 0.941 | 0.926 | 0.877 | 0.933 | 0.923 | 0.906 | 0.883 | 0.868 |
| | median | 0.987 | 0.970 | 0.922 | 0.967 | 0.969 | 0.943 | 0.935 | 0.925 |
| | mean | 0.964 | 0.945 | 0.888 | 0.961 | 0.947 | 0.922 | 0.900 | 0.887 |
| | upper quartile | 1.052 | 1.027 | 0.989 | 1.025 | 1.025 | 1.013 | 0.985 | 0.949 |

**b. Calibration under distribution shifts.** Following our observation that our uncertainty estimates are effective in rejecting outliers in 4b, here we study if they can be leveraged to calibrate ImageNet models under distribution shifts. To calibrate a classifier, we simply scale the logits of the mean by the uncertainties as follows: $\boldsymbol{\mu}_{\text{calib.}} = \boldsymbol{\mu}(1 - \bar{\boldsymbol{\sigma}})$, where $\bar{\boldsymbol{\sigma}}$ is the standard deviation estimated from (6), where once again for classification we simply compute the average of standard deviation across all the 1000 classes, followed by min-max normalization to $[0, 1)$. This simple scaling of the mean reflects our prior belief – a highly certain prediction must remain unchanged, whereas an uncertain one gets tempered down. Note, this scaling is applied to logits from all classes and hence the accuracy of the mean remains unchanged before or after calibration. We evaluate how calibrated the predictions are using three commonly used metrics – Calibration error (ECE), negative log likelihood (NLL), and Brier Score. We use the same ResNet-50 classifier trained on ImageNet as before, and measure calibration for predictions on 16 different ImageNet-C corruptions at severity 5. We list the corruptions and show examples of them in the appendix. We report the 25$^{\text{th}}$ (lower quartile), 50$^{\text{th}}$ (median) and 75$^{\text{th}}$ (upper quartile) along with the mean of the three metrics across 16 corruptions in Table 1. Across all measures we see that $\Delta-$UQ is able to calibrate models better, even in comparison to state-of-the-art approaches that use an explicit calibration objective to adjust the prediction probabilities, further validating the quality of its uncertainty estimates.

**c. Sequential Optimization.** Denoting a high-dimensional function as $f : \mathcal{D} \rightarrow \mathbb{R}$, our goal is to solve the following optimization problem: $x^* = arg \max_{x \in \mathcal{D}} f(x)$. Here, $\mathcal{D}$ refers to a *bounded* design space comprising $D$ different parameters with their corresponding value ranges $[\ell_d, h_d], \forall d = 1 \cdots D$. The high computational or financial cost of evaluating $f$ (invoking a simulator or running an experiment) motivates the additional objective of minimizing the number of evaluations.

Table 2: **Sequential optimization:** We rigorously evaluate the performance of different uncertainty estimators on a suite of black-box functions and report the AUC metric (↑) averaged across multiple random seeds and trials. In each case, we also indicate the number of initial samples and optimization steps.

| Function | Dim. | Init. | Steps | GP | MCD | BNN | DEns | Ours |
|---|---|---|---|---|---|---|---|---|
| Multi Optima | 1 | 5 | 25 | $0.51 \pm 0.2$ | $0.45 \pm 0.16$ | $0.64 \pm 0.12$ | $0.28 \pm 0.17$ | $0.73 \pm 0.09$ |
| Ackley | 2 | 5 | 25 | $0.23 \pm 0.08$ | $0.76 \pm 0.03$ | $0.71 \pm 0.1$ | $0.75 \pm 0.04$ | $0.83 \pm 0.03$ |
| Beale | 2 | 5 | 25 | $0.64 \pm 0.31$ | $0.55 \pm 0.22$ | $0.27 \pm 0.17$ | $0.81 \pm 0.03$ | $0.85 \pm 0.04$ |
| Booth | 2 | 5 | 25 | $0.39 \pm 0.21$ | $0.55 \pm 0.14$ | $0.3 \pm 0.2$ | $0.68 \pm 0.06$ | $0.79 \pm 0.04$ |
| Branin | 2 | 5 | 25 | $0.35 \pm 0.28$ | $0.28 \pm 0.19$ | $0.22 \pm 0.14$ | $0.46 \pm 0.1$ | $0.67 \pm 0.06$ |
| Bukin | 2 | 5 | 25 | $0.36 \pm 0.12$ | $0.55 \pm 0.07$ | $0.38 \pm 0.11$ | $0.59 \pm 0.11$ | $0.76 \pm 0.1$ |
| Camel | 2 | 5 | 25 | $0.83 \pm 0.08$ | $0.86 \pm 0.06$ | $0.84 \pm 0.03$ | $0.83 \pm 0.07$ | $0.89 \pm 0.03$ |
| Dropwave | 2 | 5 | 25 | $0.68 \pm 0.15$ | $0.57 \pm 0.18$ | $0.67 \pm 0.13$ | $0.67 \pm 0.11$ | $0.79 \pm 0.14$ |
| Griewank | 2 | 5 | 25 | $0.83 \pm 0.02$ | $0.74 \pm 0.04$ | $0.59 \pm 0.17$ | $0.7 \pm 0.14$ | $0.86 \pm 0.03$ |
| Holder | 2 | 5 | 25 | $0.12 \pm 0.06$ | $0.36 \pm 0.28$ | $0.36 \pm 0.37$ | $0.39 \pm 0.29$ | $0.57 \pm 0.07$ |
| Levi N.13 | 2 | 5 | 25 | $0.26 \pm 0.26$ | $0.75 \pm 0.1$ | $0.7 \pm 0.1$ | $0.6 \pm 0.11$ | $0.87 \pm 0.07$ |
| Levy | 2 | 5 | 25 | $0.57 \pm 0.18$ | $0.61 \pm 0.32$ | $0.55 \pm 0.16$ | $0.59 \pm 0.16$ | $0.83 \pm 0.03$ |
| Hartmann | 3 | 5 | 25 | $0.57 \pm 0.07$ | $0.49 \pm 0.11$ | $0.46 \pm 0.16$ | $0.53 \pm 0.15$ | $0.68 \pm 0.07$ |
| Ackley | 4 | 10 | 25 | $0.17 \pm 0.02$ | $0.06 \pm 0.04$ | $0.1 \pm 0.03$ | $0.14 \pm 0.02$ | $0.59 \pm 0.05$ |
| Griewank | 4 | 10 | 25 | $0.37 \pm 0.08$ | $0.47 \pm 0.06$ | $0.39 \pm 0.05$ | $0.43 \pm 0.07$ | $0.69 \pm 0.07$ |
| Levy | 4 | 10 | 25 | $0.1 \pm 0.08$ | $0.4 \pm 0.3$ | $0.27 \pm 0.21$ | $0.21 \pm 0.1$ | $0.62 \pm 0.2$ |
| Hartmann | 6 | 10 | 25 | $0.15 \pm 0.01$ | $0.2 \pm 0.08$ | $0.1 \pm 0.05$ | $0.15 \pm 0.04$ | $0.27 \pm 0.15$ |
| Ackley | 8 | 10 | 50 | $0.06 \pm 0.09$ | $0.09 \pm 0.13$ | $0.08 \pm 0.12$ | $0.11 \pm 0.05$ | $0.36 \pm 0.09$ |
| Griewank | 8 | 10 | 50 | $0.07 \pm 0.02$ | $0.12 \pm 0.13$ | $0.08 \pm 0.07$ | $0.19 \pm 0.07$ | $0.32 \pm 0.11$ |
| Levy | 8 | 10 | 50 | $0.12 \pm 0.04$ | $0.16 \pm 0.08$ | $0.11 \pm 0.07$ | $0.13 \pm 0.07$ | $0.47 \pm 0.03$ |
| Avg. Rank | - | - | - | 3.8 | 3.0 | 4.1 | 2.95 | 1.0 |

Given the high-dimensional nature of the design spaces, a simple brute-force search or even space-filling random sample designs [28] often require significantly large sample sizes to identify the optima, thus motivating the use of *sequential optimization* strategies. In particular, Bayesian Optimization (BO) techniques based on statistical surrogates (e.g., Gaussian processes) form an important class of solutions [29]. In a nutshell, given an initial experiment design and their function evaluations, $\{x_i, f(x_i)\}_{i=0}^{n_0}$, sequential optimization techniques incrementally select candidates to achieve the so-called *exploration-exploitation* trade-off using an appropriate *acquisition* function [30]. In this study, we use the popular expected improvement (EI) score to perform candidate selection.

*Setup*: In this experiment, we consider a large suite of black-box optimization functions with varying dimensionality (1 to 8) and complexity. We provide definitions and details of these functions used in our study in the appendix. We also perform an experiment with a pre-trained generative model (GAN) trained on MNIST handwritten digits, wherein we perform optimization in the $100-$D latent space $\mathcal{Z}$ such that thickness of the resulting digit is maximized: $\sum_i \mathbb{I}(x_i > 0), \forall i$, where $\mathbb{I}$ denotes the identity function. We use the following baseline uncertainty estimation approaches in our study: (i) Gaussian processes (GP); (ii) Monte-Carlo dropout (MCD); (iii) Bayesian neural networks (BNN) trained via variational inferencing; and (iv) deep ensembles (DEns). For all neural network surrogates, we computed positional embeddings (sinusoidal) of the raw parameter inputs prior to building a

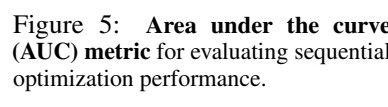

Figure 5: **Area under the curve (AUC) metric** for evaluating sequential optimization performance.

fully-connected network with 4 hidden layers each containing 128 neurons and ReLU activation. All methods were trained with the same set of hyperparameters: Adam optimizer learning rate $1e-4$ and 500 epochs, except for BNN, which required 1000 epochs for convergence. With MCD, we used

50 forward passes at test time for each sample to obtain the uncertainties. Finally, with $\Delta-$UQ, we set the number of anchors for inferencing as $\min(20, n)$, where $n$ is the number of samples in the observed dataset in any iteration.

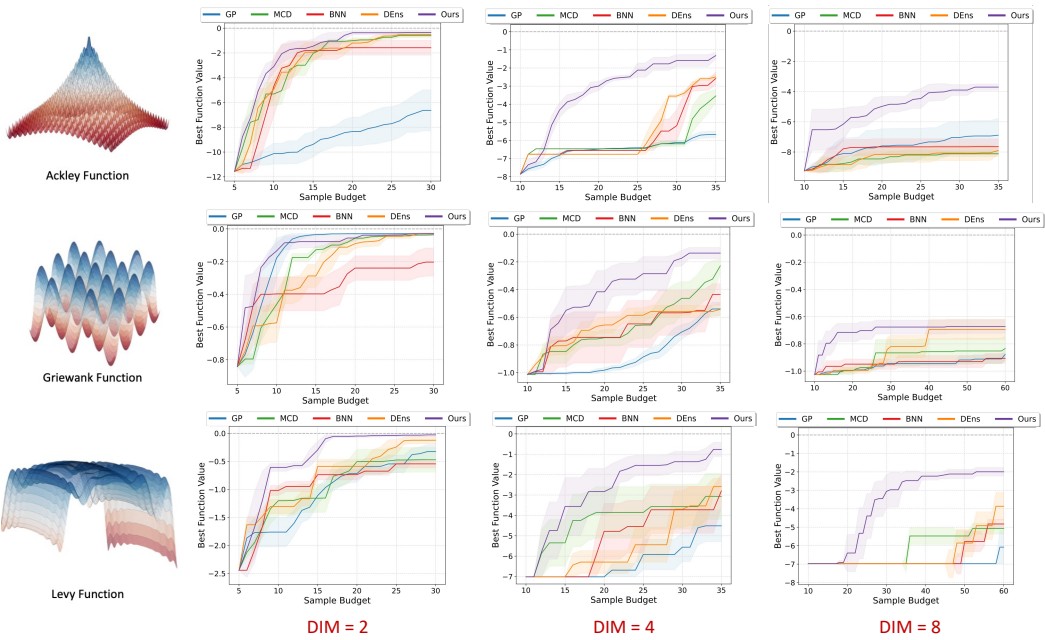

Figure 6: **Convergence curves obtained with different uncertainty estimation methods:** We show the best function value achieved for three different functions at dimensions 2, 4 and 8 respectively (for 1 random seed, 5 trials). We find that $\Delta-$UQ consistently outperforms all other baselines. The effectiveness of our approach in producing meaningful uncertainties at small sample sizes becomes more apparent as dimensionality increases.

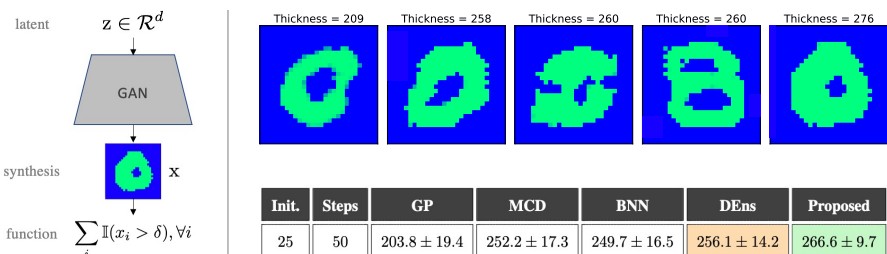

Figure 7: **GAN-based optimization**: $\Delta-$UQ consistently produces images with higher function values (thickness) for the same sampling budget, when compared to existing baseline methods.

The DEns model was constructed using 5 constituent members (increasing this did not provide any benefits), each trained with a different initialization. The number of initial samples and the number of steps in the sequential optimization were set to be the same across all methods. In each round of the Bayesian optimization, we used $10,000$ samples for initialization and 15 restarts (i.e., starting points for multistart acquisition function optimization), and finally one candidate ($q = 1$) was evaluated with the black-box function and added to the observed dataset. We performed experiments with 5 random seeds (different initializations), each for 5 independent trials. Since the goal is to reach the global optima with the fewest number of samples, we use the widely adopted area under the *iteration vs best achieved function value* curve to obtain a holistic evaluation of different approaches (see Figure 5).

*Results*: From table 2, we find that $\Delta-$UQ produces significantly higher AUC scores in comparison to existing baselines, across all benchmark functions. While MCD and DEns behave reasonably well in low dimensions, their performance suffers when we go to higher dimensions (see figure 6). Furthermore, we find that the performance of BNN is generally lower due to the inherent small samples sizes that we operate in. Finally, as shown in figure 7, our approach consistently achieves

higher function values in the MNIST GAN-based optimization, thus validating the quality of the uncertainties produced via anchoring.

## 5  Broader Impact

We presented a simple, scalable, and accurate single model uncertainty estimator that outperforms many existing techniques. Uncertainty quantification (UQ) plays a significant role in a variety of applications in science and engineering, from developing safeguards in critical applications such as healthcare, security, or finance to exploration of novel design spaces using sequential optimization. Our paper addresses both of these applications using demonstrative problems to showcase the potential capabilities. Our goal is to present a fundamental approach for uncertainty estimation that maybe applicable in different kinds of problems and settings. Due to its fundamental nature, we do not foresee misuse directly. However, we note that our method does not mitigate biases that may come with the training data, which could potentially affect the quality of uncertainties themselves.

## Acknowledgement

This work was performed under the auspices of the U.S. Department of Energy by Lawrence Livermore National Laboratory under Contract DE-AC52-07NA27344. Supported by the LDRD Program under projects 21-ERD-028, 22-ERD-006; with IM release number LLNL-JRNL-836221.

## Appendix

The appendix contains the following additional information: (a) expanded derivations, (b) PyTorch code snippets for $\Delta-$UQ training & inference; (c) results on calibration in standard regression settings on the UCI benchmarks dataset; (d) results on accuracy of predictions on ImageNet and CIFAR-10 and their corrupted variants; (e) calibration results on CIFAR10-C; (f) ablations on anchor-based training; (f) details on the design optimization experiment, including definitions for the benchmark functions and convergence plots.

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
