# Supplement: Single Model Uncertainty Estimation via Stochastic Data Centering

# APPENDIX

## A  Derivation for shifted training on NTK

We continue the derivation from the main here in more detail. Recall, the prediction on a test sample $x_t$ in the limit as the inner layer widths grow to infinity. It has been shown that (c.f. [1, 2]):

$$f_\infty(x_t) = f_0(x_t) - \mathbf{K}_{x_t \mathbf{X}} \mathbf{K}_{\mathbf{XX}}^{-1}(f_0(\mathbf{X}) - \mathbf{Y}), \tag{1}$$

where $\mathbf{X}$ is the matrix of all training data samples. As before, we consider the case where the domain is shifted by $c$. Using (1):

$$f_\infty(x_t - c) = f_0(x_t - c) - \mathbf{K}_{(x_t - c)(\mathbf{X} - c)} \mathbf{K}_{(\mathbf{X} - c)(\mathbf{X} - c)}^{-1}(f_0(\mathbf{X} - c) - \mathbf{Y})$$

$$\approx f_0(x_t - c) - (\mathbf{K}_{x_t \mathbf{X}} - \Gamma_{x_t, \mathbf{X}, c})(\mathbf{K}_{\mathbf{XX}} - \Gamma_{\mathbf{X}, \mathbf{X}, c})^{-1}(f_0(\mathbf{X} - c) - \mathbf{Y}) \tag{2}$$

Where we utilize Woodbury's Identity [3] for expanding the inverse of the difference between two matrices as:

$$(A - B)^{-1} = A^{-1} + \sum_{m=1}^{\infty}(A^{-1}B)^m A^{-1} \tag{3}$$

Using (3), we can expand (2) as:

$$= f_0(x_t - c) - (\mathbf{K}_{x_t \mathbf{X}} - \Gamma_{x_t, \mathbf{X}, c})\left(\mathbf{K}_{\mathbf{XX}}^{-1} + \sum_{m=1}^{\infty}(\mathbf{K}_{\mathbf{XX}}^{-1}\Gamma_{\mathbf{X}, \mathbf{X}, c})^m \mathbf{K}_{\mathbf{XX}}^{-1}\right)(f_0(\mathbf{X} - c) - \mathbf{Y})$$

$$\tag{4}$$

$$= f_0(x_t - c) - (\mathbf{K}_{x_t \mathbf{X}} - \Gamma_{x_t, \mathbf{X}, c})\mathbf{K}_{\mathbf{XX}}^{-1}(f_0(\mathbf{X} - c) - \mathbf{Y}) - \qquad \text{(contd.)}$$

$$(\mathbf{K}_{x_t \mathbf{X}} - \Gamma_{x_t, \mathbf{X}, c})\sum_{m=1}^{\infty}(\mathbf{K}_{\mathbf{XX}}^{-1}\Gamma_{\mathbf{X}, \mathbf{X}, c})^m \mathbf{K}_{\mathbf{XX}}^{-1}(f_0(\mathbf{X} - c) - \mathbf{Y})$$

$$= \underbrace{f_0(x_t - c) - \mathbf{K}_{x_t \mathbf{X}} \mathbf{K}_{\mathbf{XX}}^{-1}(f_0(\mathbf{X} - c) - \mathbf{Y})}_{\text{first}} - \qquad \text{(contd.)}$$

$$\Gamma_{x_t, \mathbf{X}, c}\mathbf{K}_{\mathbf{XX}}^{-1}(f_0(\mathbf{X} - c) - \mathbf{Y}) - (\mathbf{K}_{x_t \mathbf{X}} - \Gamma_{x_t, \mathbf{X}, c})\sum_{m=1}^{\infty}(\mathbf{K}_{\mathbf{XX}}^{-1}\Gamma_{\mathbf{X}, \mathbf{X}, c})^m \mathbf{K}_{\mathbf{XX}}^{-1}(f_0(\mathbf{X} - c) - \mathbf{Y})$$

$$\tag{5}$$

Next, we consider expanding the first term in (5). Since the only term dependent on $c$ is the evaluation of the network with the initial weights $\theta_0$, i.e., of the general form $f_0(x - c)$. We will expand this using a Taylor series approximation by evaluating it at $c = 0$, as following:

$$f_0(x - c) = f_0(x) + c f_0'(x) + c^2 f_0''(x) + \ldots \tag{6}$$

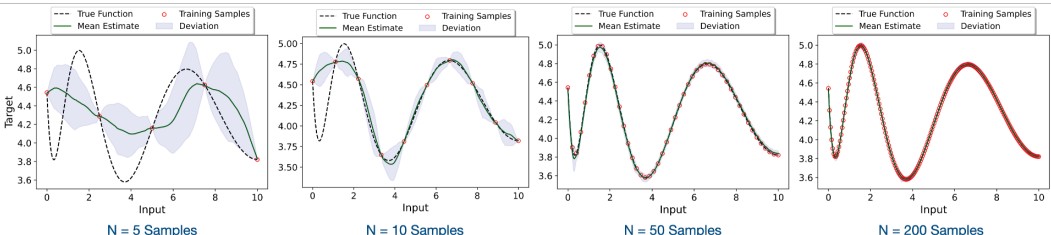

Figure 1: Behavior of the proposed uncertainty estimator as we increase the training sample size for a $1-$D regression example. As expected, as we increase $N$ from 5 samples to 200 samples, the prediction uncertainties shrink to trivial estimates, thus emphasizing the ability of our approach in capturing epistemic uncertainties.

By substituting (6) in (5), and grouping all the terms that do not depend on $c$, we can separate the deterministic and stochastic (in $c$) which gives us our final result as:

$$\approx \underbrace{f_0(x_t) - \mathbf{K}_{x_t \mathbf{X}} \mathbf{K}_{X\mathbf{X}}^{-1}(f_0(\mathbf{X}) - \mathbf{Y})}_{\text{deterministic for fixed } \boldsymbol{\theta}_0} - \underbrace{g(c, x_t, \mathbf{X}, \mathbf{Y})}_{\text{random due to c}} \qquad (7)$$

**Perturbations with different anchors**    The analysis above can be easily applied to the case where different anchors are used with different input samples. Let us consider two randomly chosen anchors: $c_1, c_2$ and study the dot product between two points shifted using these anchors.

$$[c_1, x_1 - c_1]^\top [c_2, x_2 - c_2] = x_1^\top x_2 + 2c_1^\top c_2 - c_1^\top x_2 - c_2^\top x_1 \qquad (8)$$
$$= x_1^\top x_2 - c_1^\top (c_1 c_2^\top x_1 + x_2 - 2c_2),$$

Where in the last step we exploit the fact that $c_1, c_2$ are normalized to be on the hypersphere. We can see that by setting $v = c_1 c_2^\top x_1 + x_2 - 2c_2$, we can get a similar form of perturbation as $x_1^\top x_2 - c_1^\top v$ by combining all $c_2$ related terms as before (and equivalently for $c_1$).

In this paper, we argue that the proposed stochastic data centering technique is effective at estimating epistemic uncertainties with deep networks. For demonstration, let us consider the 1D regression example showed in Figure 1 and train $\Delta-$UQ models under different train sample sizes ($5, 10, 50$ and 200 respectively). The figure illustrates the predicted function and the associated uncertainty estimates (shaded region around the predictions). As the training sample size increases, we notice that the uncertainties shrink to trivial values (very close to 0), thus validating that our estimates are strongly correlated with epistemic uncertainties.

# B    Corruptions in the anchoring process

When scaling $\Delta-$UQ to image data, especially using powerful base networks such as ResNets, we observe a consistency training significantly improves the uncertainty estimates. To achieve this, we make the input transformation less trivial as $x \to \mathcal{T}(c), x - c$, where $\mathcal{T}$ is a standard augmentation technique already used in training such as random crops or blurs. Such that, during inference we set $\mathcal{T} = \mathcal{I}$, to be identity. We apply this corruption once every 10 iterations – this is a hyper parameter, doing this more frequently makes the process much harder resulting in a worse mean estimate whereas making it less frequently results in uncertainties that are slightly worse. We outline the exact set of corruptions used in the pytorch pseudo code listed below.

## B.1    Pytorch implementation

Algorithm 1 lists the Pytorch pseduo-code for training a $\Delta-$UQ for image classification, and a sample inference script in algorithm 2. The example assumes a model as defined in algorithm 1 and a set of anchors drawn from the training distribution at random. Once predictions per anchor are obtained, the mean and standard deviation are returned as the final prediction of the model, and the corresponding uncertainty on the test samples.

**Algorithm 1** PyTorch-style example for $\Delta-$UQ with ResNet-50.

```
def create_anchored_model(model):
    model.conv1 = nn.Conv2d(in_channels=6, 64)
    return model

Tx = transforms.Compose([
    transforms.RandomResizedCrop(size=224),
    transforms.RandomHorizontalFlip(),
    transforms.RandomApply([color_jitter,blurr], p=0.8),
    ])
## load model and change the first conv layer

model_basic = ResNet50(pre_trained=False,n_class=1000)
model = create_anchored_model(model_basic)

## load datasets, setup optimizer, define criterion etc.
for i, (images, targets) in enumerate(train_loader):

    anchors = Shuffle(images)

    diff = images-anchors

    if i % 10 ==0:
        tx_anchors = Tx(anchors)
    else:
        tx_anchors = anchors

    batch = torch.cat([tx_anchors,diff],axis=1)
    output = model(batch)

    loss = criterion(output, target)

    optimizer.zero_grad()

    loss.backward()

    optimizer.step()
```

**Algorithm 2** Inference with $\Delta-$UQ for a classification model

```
'''
model       : network trained with anchoring
anchors     : set of randomly chosen anchors (ideally from train dist.)
test_inputs : samples on which predictions are needed
'''
preds = []
for A in anchors:
    D = test_inputs-A
    X_test = torch.cat([A, D],axis=1)
    y_test = model(X_test)
    preds.append(y_test)
P = torch.cat(preds,0)
mu = P.mean(0)
unc = P.std(0).sum(1) ## sum unc. along classes
```

## B.2  ImageNet-C corruptions for OOD and Calibration

Table 1 lists the set of corruptions used to construct the ImageNet-C benchmark.

Table 1: ImageNet-C corruptions used for the calibration study

| | | | |
|---|---|---|---|
| brightness | contrast | defocus_blur | elastic_transform |
| fog | frost | gaussian_blur | gaussian_noise |
| glass_blur | glass_blur | glass_blur | gaussian_noise |
| shot_noise | spatter | speckle_noise | zoom_blur |

## C  Additional Results: Prediction Performance on UCI Benchmarks

While our outlier rejection, calibration and sequential optimization experiments clearly established the effectiveness of the proposed uncertainty estimator, we also evaluate the quality of $\Delta-$UQ models, in terms of standard prediction fidelity metrics (regression in this section and classification in next).

Table 2: Regression performance evaluation using UCI benchmarks. For each case, we show the negative log-likelihood for the test data obtained using each of the methods. Note, all metrics were computed as an average from 20 random trials of $0.8 - 0.2$ train-test split. We followed the experimental setup described in [4] and the results for the baselines were obtained from the uncertainty baselines github page [5]

| Function | MCD | DEns | BNN | PBP | Proposed |
|---|---|---|---|---|---|
| Boston Housing | 2.4 | 6.11 | 3.12 | 2.54 | 2.58 |
| Concrete Strength | 2.93 | 3.2 | 3.22 | 3.04 | 3.09 |
| Energy Efficiency | 1.21 | 0.61 | 0.93 | 1.01 | 0.56 |
| Kin8nm | -1.14 | -1.17 | -1.03 | -1.28 | -1.19 |
| Naval Propulsion | -4.45 | -5.17 | -6.12 | -4.85 | -5.86 |
| Power Plant | 2.8 | 3.18 | 2.85 | 2.78 | 2.83 |
| Wine | 0.93 | 0.97 | 1.0 | 0.97 | 0.91 |
| Protein | 2.87 | 3.12 | 2.93 | 2.77 | 2.79 |
| Yacht | 1.25 | 0.73 | 2.01 | 1.64 | 0.66 |
| Avg. Rank | 2.89 | 3.56 | 4.0 | 2.44 | 2.0 |

For this study, we used a suite of regression datasets typically adopted for evaluating deep models, evaluated using the standard experiment protocol in the benchmark defined by [5]. For each of the datasets, we fit networks with a single hidden layer (50 neurons) and ReLU activation. We trained 20 independent models with different random $80 - 20$ train-test splits and report the average performance across the trials. For evaluation, we used the negative log-likelihood metric (lower the better). In addition to our approach, we include the results for MCD, DEns, BNN (variational inferencing) and Probabilistic Backpropagation [4] (with a Matrix-Variate Gaussian prior). Furthermore, for a holistic evaluation, we also report the average rank (across the 5 methods) from the suite of datasets considered. As showed in Table 2, $\Delta-$UQ performs competitively over other baselines, and achieves an average rank of $2.0$. Overall, we find that, in addition to producing high-quality uncertainty estimates, the proposed approach also produces high-quality predictive models.

## D    Additional Results: Prediction Performance on Imagenet and CIFAR-10

**ImageNet-C accuracy.** We provide results for classification accuracy of our ImageNet model on the validation set and the distribution shifted variants of ImageNet-C, in table 3. Here, at each severity level ("1"–"5") we compute the accuracy of the model across all 16 corruptions for that severity outlined in 1 and report the mean accuracy. We also report the accuracy numbers for the corresponding uncertainty baselines, and see that $\Delta-$UQ does not compromise on accuracy on the clean data, while being highly competitive to Deep Ensembles even on the most severe corruptions.

Table 3: Accuracy of ResNet-50 Model on ImageNet validation and its distribution shifted variants.

| Method | ImageNet-C Dist. Shift Variants (ResNet-50) | | | | | | |
|---|---|---|---|---|---|---|---|
| | val | 1 | 2 | 3 | 4 | 5 | Avg. |
| Vanilla | 76.1 | 62.5 | 52 | 42 | 30 | 19.5 | 47.8 |
| DEns | 78.1 | 66 | 56 | 47 | 36 | 22 | 50.05 |
| MC Dropout | 75 | 60 | 50 | 38 | 29 | 17 | 46 |
| SVI | 76.1 | 63 | 53 | 43 | 31 | 20 | 48.05 |
| $\Delta-$UQ | 76.1 | 61.7 | 53.1 | 44.2 | 33.2 | 21.8 | 48.95 |

**CIFAR-10C/ResNet-20** We perform detailed analysis of calibration and accuracy on CIFAR-10 and its corrupted variants CIFAR-10C [6] using the experimental protocol followed by [7, 8], where we use a ResNet-20 [9] and report the calibration scores across all 5 corruption levels and the validation set – the calibration metrics are reported by averaging the performance across 5 random seeds of the model. We report the average accuracy for each corruption level in table 4, and display the calibration metrics – ECE, NLL and Brier Score in figure 2. In both the accuracy and calibration metrics we find that $\Delta-$UQ outperforms all the comparable baselines, including Deep Ensembles, though using only a single model.

Table 4: Accuracy of ResNet-20 Model on CIFAR10 validation and its distribution shifted variants.

| Method | CIFAR10-C Dist. Shift Variants (ResNet-20) | | | | | | |
|---|---|---|---|---|---|---|---|
| | val | 1 | 2 | 3 | 4 | 5 | Avg. |
| Vanilla | 90.5 | 81.8 | 75.1 | 68.3 | 60.6 | 49.1 | 69.8 |
| DEns | 93.4 | 85.9 | 79.8 | 73.1 | 65 | 52.4 | 72.9 |
| MC Dropout | 91 | 83.7 | 77.5 | 70.1 | 61.5 | 49.4 | 70.2 |
| SVI | 88.6 | 82.3 | 76.9 | 70.8 | 63.1 | 52.6 | 70.6 |
| $\Delta-$UQ | 92.3 | 85.8 | 80.9 | 75.2 | 67.8 | 56.2 | 74.25 |

**Ablation studies.** We mainly perform ablation on the inference part of $\Delta-$UQ – we try to quantify the variability in performance when anchoring is not used, and when uncertainty is not used in terms of the calibration metrics for the ImageNet-C/ResNet-50 experiment considered in the main paper. We consider two main ablations of the main model as stated next. Note, in all three cases the only difference is the procedure for inference. The training procedure is kept fixed across all models, and we use the same ResNet-50 model to perform these ablation studies reported in table 5.

- **Naïve**: We consider the case where a model trained with anchoring as usual, is used during inference *without anchoring*, i.e., instead of passing $\{c, x - c\}$ as before for an anchor $c$, we pass $\{0, x\}$, as this behaves as a naive model that does not have the benefit of obtaining uncertainties or ensembling like behavior as $\Delta - UQ$.

- **Ensemble mean**: Next, we consider the version where anchoring is done during inference and compute the mean of predictions from different anchors as before, but we do not use the uncertainties obtained – i.e., the final prediction is simply the mean of the predictions obtained with different anchors.

- $\Delta-$UQ: This is our final model that takes the mean and scales it by the uncertainties during inference.

We observe from the results in table 5 first that simply training with anchoring shows benefits in model performance even if anchoring is not used during inference, as seen in improvement in the calibration performance of the naïve model (shown as $\{0, x\}$ in the table) over the vanilla model. Next, we see that using anchoring to compute the mean improves performance further as seen in the next column (shown as $\mu$), and finally factoring in the uncertainties performs the best. Even the ablated versions perform competitively compared to some of the other uncertainty baselines, indicating the effectiveness of $\Delta-$UQ .

# E  Details on GAN-based Optimization Experiment

In this experiment, we evaluated the utility of the proposed uncertainty estimator in guiding sequential optimization in the latent space of a pre-trained GAN network. We begin by assuming access to a generative model $G(z)$, which maps a latent noise vector $z$ onto a realization on the training image manifold. Denoting the latent space as $\mathcal{Z}$, our goal is to maximize a scalar function defined for an image, i.e., $f(x)$ by performing optimization in the latent space.

$$\arg \max_{z \in \mathcal{Z}} f(G(z)). \tag{9}$$

In our experiment, we used a GAN trained on MNIST hand-written images and defined the thickness function (total number of non-zero pixels in an image) for optimization. The dimensionality of the

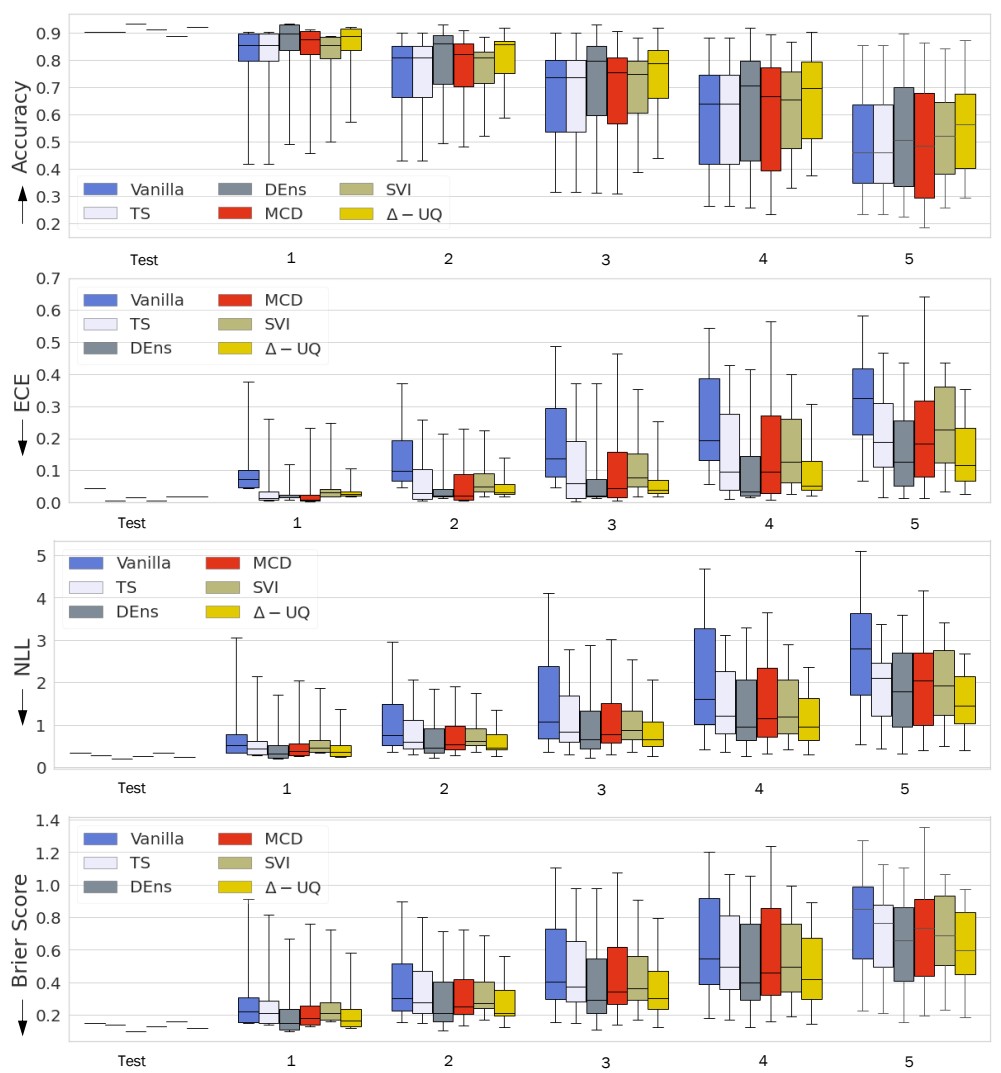

Figure 2: Calibration experiments using CIFAR-10C with a ResNet-20 model. We report average metrics for model accuracy and calibration across 5 random seeds and corruption severity levels. Note, we calibrate our predictions as before using a scaling strategy as: $\bar{\boldsymbol{\mu}} = \boldsymbol{\mu}(1 - \sigma)$. We compare against standard baselines obtained from [8, 7].

noise latent space was set to 100. Similar to our design optimization experiments with synthetic data, we started with an initial random sample (uniform random in the latent space) of 25, synthesized the corresponding images using the generator and computed the thickness function for each of them. We performed optimization for 50 steps (with 1 sample in each round) and evaluated the maximum thickness achieved as the metric of choice (the global optimum is not known). We repeated the experiment across 5 random seeds and 5 independent trials for each seed. The results in Figure 7 (main paper) illustrate the maximum thickness obtained using different uncertainty estimators across 25 experiments. We find that our approach consistently produced the highest function value and outperformed other approaches. As expected, DEns performed the second best, followed by MCD. Interestingly, with BNNs, the variational inferencing technique is known to lead to underfitting and we find out, despite achieving reasonably higher function values, the reconstructions were of significantly poorer quality (off the image manifold).

Table 5: **Calibration Comparison With Ablation:** We study how different ablations of our model perform on the calibration task.

| Metric | | Vanilla | Temp Scaling | DEns | MCD | SVI-AvUC | $\{0, \mathrm{x}\}$ | $(\boldsymbol{\mu})$ | $\Delta-\mathrm{UQ}$ |
|---|---|---|---|---|---|---|---|---|---|
| ECE ↓ | lower quartile | 0.124 | 0.096 | 0.050 | 0.078 | 0.032 | 0.077 | 0.085 | 0.022 |
| | median | 0.174 | 0.139 | 0.090 | 0.134 | 0.045 | 0.117 | 0.112 | 0.038 |
| | mean | 0.194 | 0.160 | 0.088 | 0.153 | 0.054 | 0.130 | 0.110 | 0.044 |
| | upper quartile | 0.274 | 0.236 | 0.126 | 0.219 | 0.070 | 0.193 | 0.125 | 0.063 |
| NLL ↓ | lower quartile | 4.635 | 4.53 | 4.035 | 4.699 | 4.164 | 4.011 | 4.072 | 4.014 |
| | median | 5.115 | 4.993 | 4.624 | 5.093 | 4.823 | 4.818 | 4.679 | 4.617 |
| | mean | 5.234 | 5.091 | 4.604 | 5.553 | 4.707 | 4.832 | 4.516 | 4.352 |
| | upper quartile | 6.292 | 6.165 | 5.893 | 6.522 | 5.778 | 5.925 | 5.124 | 4.987 |
| Brier ↓ | lower quartile | 0.941 | 0.926 | 0.877 | 0.933 | 0.883 | 0.882 | 0.887 | 0.868 |
| | median | 0.987 | 0.970 | 0.922 | 0.967 | 0.935 | 0.944 | 0.940 | 0.925 |
| | mean | 0.964 | 0.945 | 0.888 | 0.961 | 0.900 | 0.926 | 0.903 | 0.887 |
| | upper quartile | 1.052 | 1.027 | 0.989 | 1.025 | 0.985 | 1.026 | 0.972 | 0.949 |

# F   Benchmark Functions

Figure 3 illustrates the different benchmark functions for evaluating the proposed approach in black-box optimization. In addition to the functions listed, we also considered the Hartmann functions, in dimensions 3 and 6 respectively, defined as follows.

**Hartmann3:** $f(\mathrm{x}) = \sum_{i=1}^{4} \alpha_i \exp\left( -\sum_{j=1}^{3} A_{ij}(x_j - P_{ij})^2 \right)$, where

$$\alpha = (1.0, 1.2, 3.0, 3.2)^{\top}$$

$$\mathbf{A} = \begin{pmatrix} 3.0 & 10 & 30 \\ 0.1 & 10 & 35 \\ 3.0 & 10 & 30 \\ 0.1 & 10 & 35 \end{pmatrix}, \quad \mathbf{P} = 10^{-4} \begin{pmatrix} 3689 & 1170 & 2673 \\ 4699 & 4387 & 7470 \\ 1091 & 8732 & 5547 \\ 381 & 5743 & 8828 \end{pmatrix} \tag{10}$$

**Hartmann6:** $f(\mathrm{x}) = \sum_{i=1}^{4} \alpha_i \exp\left( -\sum_{j=1}^{6} A_{ij}(x_j - P_{ij})^2 \right)$, where

$$\alpha = (1.0, 1.2, 3.0, 3.2)^{\top}$$

$$\mathbf{A} = \begin{pmatrix} 10 & 3 & 17 & 3.50 & 1.7 & 8 \\ 0.05 & 10 & 17 & 0.1 & 8 & 4 \\ 3.0 & 3.5 & 1.7 & 10 & 17 & 8 \\ 17 & 8 & 0.05 & 10 & 0.1 & 14 \end{pmatrix},$$

$$\mathbf{P} = 10^{-4} \begin{pmatrix} 1312 & 1696 & 5569 & 124 & 8283 & 5886 \\ 2329 & 4135 & 8307 & 3736 & 1004 & 9991 \\ 2348 & 1451 & 3511 & 2883 & 3047 & 6650 \\ 4047 & 8828 & 8732 & 5743 & 1091 & 381 \end{pmatrix} \tag{11}$$

# G   Detailed Results for Sequential Optimization

At the core of AI-powered applications in science and engineering lies the need to perform design optimization for maximizing a chosen target objective, and to enable automated exploration in high-dimensional parameter spaces. When $f$ is Lipschitz continuous, i.e., $\|f(\mathrm{x}) - f(\mathrm{x}')\| \leq c\|\mathrm{x} - \mathrm{x}'\|$, and its first- and second-order information are accessible, this can be solved using first-order optimization methods such as stochastic gradient descent (SGD) [10] or second-order methods such as L-BFGS [11]. However, in practice, while $f$ can be explicitly evaluated for any

| Function | Bounds | Definition |
|---|---|---|
| Multi Optima (1D) | [-1, 2] | $f(\mathrm{x}) = \sin(\mathrm{x})\cos(5\mathrm{x})\cos(22\mathrm{x})$ |
| Ackley | [-10, 10] | $f(\mathrm{x}) = a\exp\left(-b\sqrt{\frac{1}{d}\sum_{i=1}^{d}x_i^2}\right) + \exp\left(\frac{1}{d}\sum_{i=1}^{d}\cos(cx_i)\right) - a - \exp(1)$ 
 $a = 20, b = 0.2, c = 2\pi$ |
| Beale (2D) | [-4.5, 4.5] | $f(\mathrm{x}) = -(1.5 - x_1 + x_1 x_2)^2 - (2.25 - x_1 + x_1 x_2^2)^2 - (2.625 - x_1 + x_1 x_2^3)^2$ |
| Booth (2D) | [-10, 10] | $f(\mathrm{x}) = -(x_1 + 2x_2 - 7)^2 - (2x_1 + x_2 - 5)^2$ |
| Branin (2D) | [-5, 10] [0, 15] | $f(\mathrm{x}) = -a(x_2 - bx_1^2 + cx_1 - r)^2 - s(1 - t)\cos(x_1) - s$ 
 $a = 1, b = 5.1/(4\pi^2), c = 5/\pi, r = 6, s = 10, t = 1/(8\pi)$ |
| Bukin (2D) | [-15, -5] [-3, 3] | $f(\mathrm{x}) = 100\sqrt{|x_2 - 0.01x_1^2|} + 0.01|x_1 + 10|$ |
| Six-Hump Camel (2D) | [-3, 3] [-2, 2] | $f(\mathrm{x}) = -\left(4 - 2.1x_1^2 + \frac{x_1^4}{3}\right)x_1^2 - x_1 x_2 - (-4 + 4x_2^2)x_2^2$ |
| Dropwave (2D) | [-5.1, 5.1] | $f(\mathrm{x}) = \frac{1 + \cos(12\sqrt{x_1^2 + x_2^2})}{0.5(x_1^2 + x_2^2) + 2}$ |
| Griewank | [-10, 10] | $f(\mathrm{x}) = -\sum_{i=1}^{d}\frac{x_i^2}{4000} + \sum_{i=1}^{d}\cos(\frac{x_i}{\sqrt{i}}) - 1$ |
| Holder Table (2D) | [-10, 10] | $f(\mathrm{x}) = \left|\sin x_1 \cos x_2 \exp\left|1 - \frac{\sqrt{x_1^2 + x_2^2}}{\pi}\right|\right|$ |
| Levy N.13 (2D) | [-10, 10] | $f(\mathrm{x}) = -\sin^2(3\pi x_1) - (x_1 - 1)^2[1 + \sin^2(3\pi x_2)] - (x_2 - 1)^2[1 + \sin^2(2\pi x_2)]$ |
| Levy | [-10, 10] | $f(\mathrm{x}) = -\sin^2(\pi w_1) - \sum_{i=1}^{d-1}(w_i - 1)^2[1 + 10\sin^2(\pi w_i + 1)] - (w_d - 1)^2[1 + \sin^2(2\pi w_d)]$ 
 $w_i = 1 + \frac{x_i - 1}{4}, i = 1, \cdots, d$ |

Figure 3: Benchmark functions used in this paper to evaluate sequential optimization

x, its first- and second-order information are unknown, thus making such an optimization very challenging. Commonly referred to as *black-box optimization* [12], this formulation is adopted in applications ranging from drug design [13] to additive manufacturing [14] and optimizing financial investments [15] to hyper-parameter tuning in neural networks [16].

$$\mathrm{x}^* = arg\max_{\mathrm{x}\in\mathcal{D}} \mathrm{f}(\mathrm{x}). \tag{12}$$

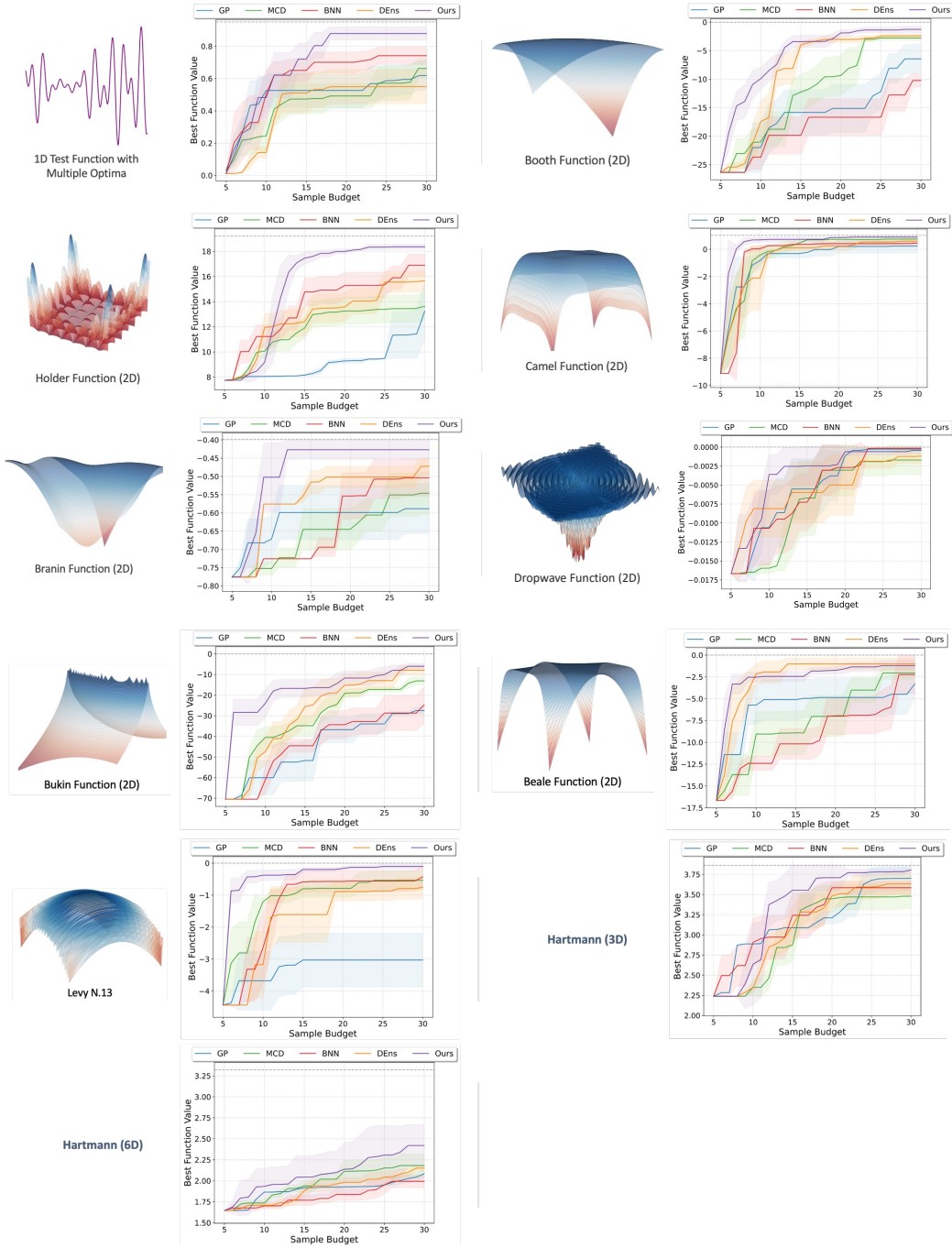

Figure 4: Convergence curves for each of the benchmark functions used in our evaluations.

In each step of this optimization, we approximate the function $f$ using the samples observed so far to obtain the surrogate $\hat{f}$. Assuming that our goal is to maximize $f$, one can acquire more samples in regimes of $\mathcal{D}$ where the mean estimate from the surrogate is high (*exploitation*) or the uncertainty is large (*exploration*). In order to balance between these two objectives and to guide the progressive search for the optima, BO utilizes an appropriate *acquisition* function [17]. In this study, we use the

popular expected improvement (EI) score to perform candidate selection.

$$\text{aq}_{\text{EI}} := \begin{cases} (\mu(\text{x}) - f(\text{x}^+) - \xi)\Phi(\text{Z}) + \sigma(\text{x})\phi(\text{Z}) & \text{if } \sigma(\text{x}) > 0 \\ 0 & \text{if } \sigma(\text{x}) = 0 \end{cases}$$

$$\text{where } \text{Z} = \frac{(\mu(\text{x}) - f(\text{x}^+) - \xi)}{\sigma(\text{x})}. \tag{13}$$

Here, $\mu(\text{x})$ and $\sigma(\text{x})$ are the mean and uncertainty estimates from the surrogate $\hat{f}$ for any sample, and $f(\text{x}^+)$ is the best known function value so far during any iteration of the optimization. Further, $\Phi(.)$ and $\phi(.)$ denote the cumulative distribution and probability density functions corresponding to the normal distribution. Finally, the hyper-parameter $\xi$ controls the exploration-exploitation trade-off. Figure 4 shows the convergence curves for each of the black-box functions.