# OpenReview forum: "Single Model Uncertainty Estimation via Stochastic Data Centering"
_NeurIPS.cc/2022/Conference — NeurIPS 2022 Accept_

### Official Review · Reviewer_DQAh · 2022-07-11

**Rating:** 6
**Confidence:** 3
**Soundness:** 3 good
**Presentation:** 3 good
**Contribution:** 3 good

**Summary:**

The authors propose $\Delta-$UQ, an approach inspired by the observation that difference in predictions from neural networks trained on constant-bias shifted data can act as strong indicators of epistemic uncertainty. For a random bias (anchor), it is shown that an "anchor ensemble" can be approximated using a single model. When evaluated on Bayesian optimization benchmarks, the authors demonstrate competitve performance hinting that the uncertainties learned are meaningful.

**Questions:**

* Is there a consistency result for $\Delta-$UQ that says that in the limit of data, uncertainty goes to zero? This would only be true for epistemic uncertainty, which appears to be the focus of uncertainty quantification in the main text. If not, then I think the distinction of aleatoric/epistemic should just be removed from the main text since it does not really add value and probably just talk about the merits of predictive uncertainty that the method provides.

* As noted earlier, can we look at the raw performance numbers for the experiments in Table 1?

**Limitations:**

Yes.

**Strengths And Weaknesses:**

Strengths:

* The authors take a different perspective to ensemble construction. Instead of working with samples of neural network parameters, they work with randomized constant-shifted datasets. The ability construct this ensemble using a single model is very practically appealing.

Weaknesses:

The authors aim to solve for well-calibrated estimate of epistemic uncertainty, which would imply uncertainty in the model definitely by parameters $\theta$. However, at test time, the authors seem to marginalize over $c$ for a given $\theta^\star$ instead of marginalizing over $\theta$. There is nothing inherently wrong with this, but it is a subtly different notion of uncertainty than what the competing methods provide. In that sense, the methods may not be entirely comparable.

Alongside the calibration metrics, it will be important to take a look at the performance numbers. Often times, calibration and predictive performance can be at odds and it will be helpful for the reader to understand what kind of numerical trade-off to expect.

Overall, I think this is technically sound and a promising direction to investigate further.

Minor:

* The GAN experiments could use a more elaborate description

---

> ### Author Response · Authors · 2022-08-01
> **Response to Reviewer DQAh**
>
> We thank the reviewer for the positive feedback and suggestions for improving the paper.
>
> **Marginalizing over c instead of \theta... may not be entirely comparable**
>
> We agree with the reviewer that the kind of ensembling achieved here is slightly different, however as we see from Figure 1 (main paper), and our NTK analysis the _effective_ NTK of the chosen network appears to be perturbed. In other words, we are able to sample from the hypothesis space by marginalizing over ‘c’ itself. We agree a deeper insight into this behavior would be interesting, which we plan to pursue as part of our future work.
>
> **GAN experiments could use a more elaborate description**
>
> Thank you for pointing this out -- we have elaborated on this experiment further in the supplement (Section 7), to include more details on how this experiment was carried out.
>
> **Is there a consistency result for UQ that says that in the limit of data, uncertainty goes to zero?**
>
> This is an interesting question -- from our empirical studies, we do observe that the uncertainty goes to zero around training points, while remaining high in unobserved data regimes. We illustrate this in a new supplement Figure 1 where we show the approximation for the same function with an increasing number of training samples. We observe that the uncertainty consistently decreases as the function fit improves -- and importantly is always zero around training samples.
>
> **Calibration and predictive performance can be at odds... can we look at the raw performance numbers for the experiments in Table 1?**
>
> We have reported accuracy scores for ImageNet as well as adding calibration, and accuracy scores for CIFAR10 with ResNet20 in the supplement. We see that in terms of accuracy our model does not compromise, and is often better than other baselines (single model uncertainty estimators) under distribution shifted variants. We reproduce the tables here for convenience.
>
> 1. Results for NLL, Brier Score and ECE for CIFAR-10 in distribution and all shifted variants are added as a new figure in the supplement (Table 4, Figure  2). We see that $\Delta$-UQ performs better than Deep Ensembles nearly in all OOD settings, and comparably with in-distribution validation data. Here is the summary table for reference.
>
> **CIFAR-10 → CIFAR-10 validation, CIFAR-10C (varying corruption severity)**
>
> | **Method**  | **val**  | **1**    | **2**    | **3**    | **4**    | **5**    | **Avg**   |
> |-------------|----------|----------|----------|----------|----------|----------|-----------|
> | vanilla     |   90.5   |   81.8   |   75.1   |   68.3   |   60.6   |   49.1   |    69.8   |
> | MCD         |    91    |   83.7   |   77.5   |   70.1   |   61.5   |   49.4   |    70.2   |
> | SVI         |   88.6   |   82.3   |   76.9   |   70.8   |   63.1   |   52.6   |    70.6   |
> | DEns        | **93.4** | **85.9** |  _79.8_  |  _73.1_  |   _65_   |  _52.4_  |   _72.9_  |
> | $\Delta-$UQ |  _92.3_ | **85.8** | **80.9** | **75.2** | **67.8** | **56.2** | **74.25** |
>
> 2. For Imagenet, we followed the protocol from [1,2] (Table 3 in the supplement). Note, we could not obtain the NLL, Brier or ECE scores for in-distribution data from the official repository, other than in the summary plots provided in their papers. We have contacted the authors, and will include the results in the camera ready version.
>
> **ImageNet → ImageNet Val, ImageNet-C (varying corruption severity)**
>
> | **Method**  | **val**  | **1**  | **2**  | **3**  | **4**  | **5**    | **Avg**   |
> |-------------|----------|--------|--------|--------|--------|----------|-----------|
> | vanilla     |  _76.1_  |  62.5  |   52   |   42   |   30   |   19.5   |    47.8   |
> | MCD         |    75    |   60   |   50   |   38   |   29   |    17    |     46    |
> | SVI         |  _76.1_  |  _63_  |   53   |   43   |   31   |    20    |   48.05   |
> | DEns        | **78.1** | **66** | **56** | **47** | **36** |  **22**  | **50.05** |
> | $\Delta-$UQ |  _76.1_  |  61.7  | _53.1_ | _44.2_ | _33.2_ | **21.8** |  _48.95_ |
>
> Response continued in the next comment.

---

> > ### Author Response · Authors · 2022-08-01
> > **Part 2: Response to Reviewer DQAh**
> >
> > 3. For UCI benchmarks, we follow the protocol from https://github.com/google/uncertainty-baselines/tree/main/baselines/uci and report the NLL metric for all cases (Table 2 in the Supplement). Here is the table for reference.
> >
> > **UCI Benchmark Regression Performance Evaluation**
> >
> > |  **UCI Dataset** |  **MCD** | **DEns** | **BNN-VI** | **PBP-MV** | **$\Delta-$UQ** |
> > |:----------------:|:--------:|:--------:|:----------:|:----------:|:-------------:|
> > | Boston           |  **2.4** |   6.11   |    3.12    |   _2.54_   |      2.58     |
> > | Concrete         | **2.93** |    3.2   |    3.22    |   _3.04_   |      3.09     |
> > | Energy           |   1.21   |  _0.61_  |    0.93    |    1.01    |    **0.56**   |
> > | Kin8nm           |   -1.14  |   -1.17  |    -1.03   |  **-1.28** |    _-1.19_    |
> > | Naval Propulsion |   -4.45  |   -5.17  |  **-6.12** |    -4.85   |    _-5.86_    |
> > | Power Plant      |   _2.8_  |   3.18   |    2.85    |  **2.78**  |      2.83     |
> > | Wine             |  _0.93_  |   0.97   |      1     |    0.97    |    **0.91**   |
> > | Protein          |   2.87   |   3.12   |    2.93    |  **2.77**  |     _2.79_    |
> > | Yacht            |   1.25   |  _0.73_  |    2.01    |    1.64    |    **0.66**   |
> > | Avg. Rank        |   2.89   |   3.56   |      4     |   _2.44_   |     **2**     |
> >
> > In case our answers have justifiably addressed your concerns, we respectfully request that you increase your score to support the acceptance of our work.

---

> > > ### Comment · Reviewer_DQAh · 2022-08-03
> > > **Response to Author Rebuttal**
> > >
> > > Thank you for your response and updated draft. I will be voting for acceptance.

---

> > > > ### Author Response · Authors · 2022-08-09
> > > > **Thank You!**
> > > >
> > > > Thank you for going over the rebuttal and voting for the acceptance of this work!

---

### Official Review · Reviewer_Gnzc · 2022-07-11

**Rating:** 5
**Confidence:** 3
**Soundness:** 2 fair
**Presentation:** 3 good
**Contribution:** 2 fair

**Summary:**

The authors propose to learn an ensemble of networks sharing the same initialisations on data set shifted by a constant. The shifting constant is augmented to the model's input.

The authors provide experiments on ImageNet dataset using ResNet50, experiments with sequential optimisation and GAN.

**Questions:**


Fig 1: It appears that the $\Delta - UQ$ is more overconfident on left hand side than the second fit?

Does the proposed method achieve better likelihood than standard deep ensembles (the results reported in Simple and Scalable Predictive Uncertainty Estimation using Deep Ensembles Lakshminarayanan et al.)

It is not clear at all how authors sample c, ad if this has been tuned on the actual test sets of the experiment (were the experiments run multiple times?)


**Limitations:**

The authors do not discuss if the method performs better or worse in terms of standard metrics and focus only on uncertainty metrics. The authors do not provide examples or insights were the method can fail.

**Strengths And Weaknesses:**

The authors propose to learn an ensemble of networks sharing the same initialisations on data set shifted by a constant. The shifting constant is augmented to the model's input.

The authors provide experiments on ImageNet dataset using ResNet50, experiments with sequential optimisation and GAN.


Like authors say, the proposed method is simple. This idea has not been considered in the literature to the best of my knowledge, but in spirit is similar to data augmentation. In a more generic way, one could learn an ensemble of networks with different data augmentations, and one would expect this increases variability of networks' predictions.

As the proposed method is simple, this paper attempts to put emphasis on the experimental side.
While the experiments are interesting, I think the results are not clearly demonstrating that the proposed method is superior.

First, I'd like to see the likelihoods of the different algorithms compared. While the method might have better properties in terms of epistemic uncertainty, likelihood on held-out data remains a standard way to compare quality of predictions.

Second, the results in a sequential optimisation experiment are mostly not statistically significant (the error bars are overlapping). It's well-known that BNNs are heavily underfitting the data and don't work well for sequential optimisation, yet it's performance in the table is sometimes close to $\Delta UQ$ (e.g. Hartmann).

The paper is mostly clear, but leaves some important experimental details not sufficiently discussed (e.g. selection of sampling distribution of c and if it is tuned).

---

> ### Author Response · Authors · 2022-08-01
> **Response to Reviewer Gnzc**
>
> We thank the reviewer for all the insightful comments and suggestions for improving the exposition of $\Delta-$UQ's performance.
>
> **Likelihoods of different algorithms compared.**
>
> Thanks for this comment. Following the reviewer’s suggestion, we have now included prediction performance for ImageNet, CIFAR and UCI benchmarks in the supplement. We summarize them below for convenience. We note that our method consistently performs better than other baselines for out of distribution data. For in distribution, data we perform similarly to Deep Ensembles. We are consistently better than all other uncertainty estimators considered here.
>
> 1. Results for NLL, Brier Score and ECE for CIFAR-10 in distribution and all shifted variants are added as a new figure in the supplement (Table 4, Figure  2). We see that $\Delta-$UQ performs better than Deep Ensembles nearly in all OOD settings, and comparably with in-distribution validation data. Here is the summary table for reference.
>
> **CIFAR-10 → CIFAR-10 validation, CIFAR-10C (varying corruption severity)**
>
> | **Method**  | **val**  | **1**    | **2**    | **3**    | **4**    | **5**    | **Avg**   |
> |-------------|----------|----------|----------|----------|----------|----------|-----------|
> | vanilla     |   90.5   |   81.8   |   75.1   |   68.3   |   60.6   |   49.1   |    69.8   |
> | MCD         |    91    |   83.7   |   77.5   |   70.1   |   61.5   |   49.4   |    70.2   |
> | SVI         |   88.6   |   82.3   |   76.9   |   70.8   |   63.1   |   52.6   |    70.6   |
> | DEns        | **93.4** | **85.9** |  _79.8_  |  _73.1_  |   _65_   |  _52.4_  |   _72.9_  |
> | $\Delta-$UQ |  _92.3_ | **85.8** | **80.9** | **75.2** | **67.8** | **56.2** | **74.25** |
>
> 2. For Imagenet, we followed the protocol from [1,2] (Table 3 in the supplement). Note, we could not obtain the NLL, Brier or ECE scores for in-distribution data from the official repository, other than in the summary plots provided in their papers. We have contacted the authors, and will include the results in the camera ready version.
>
> **ImageNet → ImageNet Val, ImageNet-C (varying corruption severity)**
>
> | **Method**  | **val**  | **1**  | **2**  | **3**  | **4**  | **5**    | **Avg**   |
> |-------------|----------|--------|--------|--------|--------|----------|-----------|
> | vanilla     |  _76.1_  |  62.5  |   52   |   42   |   30   |   19.5   |    47.8   |
> | MCD         |    75    |   60   |   50   |   38   |   29   |    17    |     46    |
> | SVI         |  _76.1_  |  _63_  |   53   |   43   |   31   |    20    |   48.05   |
> | DEns        | **78.1** | **66** | **56** | **47** | **36** |  **22**  | **50.05** |
> | $\Delta-$UQ |  _76.1_  |  61.7  | _53.1_ | _44.2_ | _33.2_ | **21.8** |  _48.95_ |
>
> 3.. For UCI benchmarks, we follow the protocol from https://github.com/google/uncertainty-baselines/tree/main/baselines/uci and report the NLL metric for all cases (Table 2 in the Supplement). Here is the table for reference.
>
> **UCI Benchmark Regression Performance Evaluation**
>
> |  **UCI Dataset** |  **MCD** | **DEns** | **BNN-VI** | **PBP-MV** | **$\Delta-$UQ** |
> |:----------------:|:--------:|:--------:|:----------:|:----------:|:-------------:|
> | Boston           |  **2.4** |   6.11   |    3.12    |   _2.54_   |      2.58     |
> | Concrete         | **2.93** |    3.2   |    3.22    |   _3.04_   |      3.09     |
> | Energy           |   1.21   |  _0.61_  |    0.93    |    1.01    |    **0.56**   |
> | Kin8nm           |   -1.14  |   -1.17  |    -1.03   |  **-1.28** |    _-1.19_    |
> | Naval Propulsion |   -4.45  |   -5.17  |  **-6.12** |    -4.85   |    _-5.86_    |
> | Power Plant      |   _2.8_  |   3.18   |    2.85    |  **2.78**  |      2.83     |
> | Wine             |  _0.93_  |   0.97   |      1     |    0.97    |    **0.91**   |
> | Protein          |   2.87   |   3.12   |    2.93    |  **2.77**  |     _2.79_    |
> | Yacht            |   1.25   |  _0.73_  |    2.01    |    1.64    |    **0.66**   |
> | Avg. Rank        |   2.89   |   3.56   |      4     |   _2.44_   |     **2**     |
>
> **The results in a sequential optimisation experiment are mostly not statistically significant**
>
> We thank the reviewer for pointing this out -- in fact the value pointed out by the reviewer is an unfortunate typo -- the AUC score of 0.6 for Hartmann-3 was supposed to be “0.46” (hence we correctly did not highlight it as second best). Our method (and even other approaches such as MCD and DEns) is significantly better than BNN consistently. We also show a comparison in terms of rank of best performing methods -- ours is ranked the best on average across all functions, whereas BNN is the worst (lower than  even plain GP), as expected due to the underfitting behavior.
>
> Response continued in the next comment.

---

> > ### Author Response · Authors · 2022-08-01
> > **Part 2: Response to Reviewer Gnzc**
> >
> > **Fig 1: It appears that the \Delta-UQ is more overconfident on left hand side than the second fit?**
> >
> > This is an accurate observation -- the example is Fig 1 shows two aspects (a) $\Delta$-UQ approximates epistemic uncertainties similar to anchor ensembling, but with a single model (computationally efficient); (b) sometimes this can come at reduced diversity compared to anchor ensembling, leading to more confident predictions than expected -- this can also be seen in the spectra of our NTK perturbation plots in Figures in Fig 1. However, in practice, these approximate uncertainties are still useful for challenging tasks such as sequential optimization.
> >
> > **It is not clear at all how authors sample c, and if this has been tuned on the actual test sets of the experiment (were the experiments run multiple times?)**
> >
> > The choice of ‘c’ is always kept random -- we do not tune this in any manner. Both during train and test the anchors are always chosen at random from the training set. During inference we simply take a single random batch (10-20 anchors) from the training set to obtain predictions on test samples. Note that, the performance does not vary significantly for different anchor choices (repeated multiple times).
> >
> > **The authors do not discuss if the method performs better or worse in terms of standard metrics and focus only on uncertainty metrics. The authors do not provide examples or insights were the method can fail.**
> >
> > We have added accuracies and likelihood numbers for ImageNet, CIFAR and UCI benchmarks in the supplement following the reviewer’s comment. In terms of failure, we have added a new ablation on variation of performance in Table 5 of the supplement to better understand how data centering impacts calibration performance on ImageNet.
> >
> > In case our answers have justifiably addressed your concerns, we respectfully request that you increase your score to support the acceptance of our work.

---

> ### Author Response · Authors · 2022-08-08
> **Gentle Reminder**
>
> Dear Reviewer Gnzc,
>
> We hope that we have justifiably addressed all your comments from your initial review. Since the discussion period ends soon, please let us know if you have any further questions. Thank you!

---

> > ### Comment · Reviewer_Gnzc · 2022-08-08
> > **Thanks for the clarification**
> >
> > I have reviewed the discussion, read the paper again and updated my score. The aggregated empirical evidence in discussion & paper convinces me to recommend borderline accept.

---

> > > ### Author Response · Authors · 2022-08-09
> > > **Thank You!**
> > >
> > > Thank you for going over the rebuttal and updating your score. Since there is one more day left for the discussions, are there any specific questions that we can address?

---

### Official Review · Reviewer_zA8F · 2022-07-12

**Rating:** 7
**Confidence:** 4
**Soundness:** 3 good
**Presentation:** 3 good
**Contribution:** 3 good

**Summary:**

This work presents $\Delta$-UQ, a simple approach to quantify deep neural network model's predictive uncertainty. The method proceeds by train a single model with constant-shift data augmentation, and produce predictive uncertainty by averaging the prediction over multiple versions of constant-shifted testing examples.

Authors conducted NTK analysis of the neural model under constant shifts, showing that at asymptotic limit, the constant-shifted model indeed lead to a different kernel. They also conducted comprehensive evaluation of the method under outlier rejection, calibration under distribution shifts, and sequential optimization, showing improved performance across all scenarios.



**Questions:**

 * Effectiveness of training-time augmentation.
   The model training procedure effectively applies a constant-shift data augmentation during training, and the inference procedure effectively applies a test-time augmentation. To this end, there exists training-time augmentation methods (e.g., AugMix) that only applies training-time augmentation, illustrating good performance. Therefore it might be interesting to decouple the contribution of the two component of the methods by studying their ablations. For example, what is the performance of a model that only receives constant-shift augmentation at training time, and at test time we only evaluate it using $[0, x]$?

**Limitations:**

The only limitation I can see is the author can consider testing it on more data modalities (e.g., UCI datasets). Given the simplicity and the generality of this approach, not illustrating its usefulness on modalities where high-quality data augmentation is not available seems to be an missing opportunity. With that said, I believe the experiments in the current paper is already sufficient to justify publication.

**Strengths And Weaknesses:**

**Strength**

 * A novel discovery that stochastic data centering is sufficient in producing effective predictive uncertainty.
 * A clear NTK analysis that strong motivates the approach, and elucidates it's impact on model's output uncertainty.
 * The method is simple to implement, and performs strongly on realistic benchmarks.

**Weakness**

I am largely happy with the paper's novelty, clarity, and evaluation, and find the finding significant. I only have a minor suggestion on conducting some ablation studies to further elucidate the contribution of different components, which I listed in **Question**

---

> ### Author Response · Authors · 2022-08-01
> **Response to Reviewer zA8F**
>
> Thank you for your kind review. We answer your questions in detail below.
>
> **what is the performance of a model that only receives constant-shift augmentation at training time, and at test time we only evaluate it using [0,x]?**
>
> Following your suggestion, we have now included calibration experiments with the following two ablated versions of our approach (note these are shown with the same model trained with anchoring, differing only in how inferencing is performed):
>
> a. $[0,x]$ model whose inference is performed by passing [0,x] for a test sample ‘x’ and therefore does not have uncertainties available.
>
> b. $\mu$ model where we do not factor uncertainties into the calibration and only look at the mean during prediction.
>
> We report these results in new Table 5 of the supplementary material. We reproduce it below for convenience. We see that simply training the model with anchoring is sufficient to improve its calibration performance over a vanilla model. Next, we see that performing anchoring during inferencing further improves performance as seen by results of the $\mu$ model.
>
> Finally, factoring in the uncertainties further improves the model as seen by $\Delta-$UQ.
> | **Metric** | **Summary** | **[0,x]** | **$\mu$** | **$\Delta-$UQ** |
> |------------|-------------|-----------|-----------|-----------------|
> | ECE        | lower       |   0.077   |   0.085   |    **0.022**    |
> |            | median      |   0.117   |   0.112   |    **0.038**    |
> |            | mean        |   0.130   |   0.110   |    **0.044**    |
> |            | upper       |   0.193   |   0.125   |    **0.063**    |
> | NLL        | lower       |   4.011   |   4.072   |    **4.014**    |
> |            | median      |   4.818   |   4.679   |    **4.617**    |
> |            | mean        |   4.832   |   4.516   |    **4.352**    |
> |            | upper       |   5.925   |   5.124   |    **4.987**    |
> | Brier      | lower       |   0.882   |   0.887   |    **0.868**    |
> |            | median      |   0.944   |   0.940   |    **0.925**    |
> |            | mean        |   0.926   |   0.903   |    **0.887**    |
> |            | upper       |   1.026   |   0.972   |    **0.949**    |
>
>
> **Testing it on more data modalities (e.g. UCI)**
>
> We have added a new table to show the negative log likelihoods for different uncertainty estimators when fitting datasets on the UCI benchmark. We see strong performance from the $\Delta-$UQ model (Details are in new Table 2 of the supplement):
>
> |  **UCI Dataset** |  **MCD** | **DEns** | **BNN-VI** | **PBP-MV** | **$\Delta-$UQ** |
> |:----------------:|:--------:|:--------:|:----------:|:----------:|:-------------:|
> | Boston           |  **2.4** |   6.11   |    3.12    |   _2.54_   |      2.58     |
> | Concrete         | **2.93** |    3.2   |    3.22    |   _3.04_   |      3.09     |
> | Energy           |   1.21   |  _0.61_  |    0.93    |    1.01    |    **0.56**   |
> | Kin8nm           |   -1.14  |   -1.17  |    -1.03   |  **-1.28** |    _-1.19_    |
> | Naval Propulsion |   -4.45  |   -5.17  |  **-6.12** |    -4.85   |    _-5.86_    |
> | Power Plant      |   _2.8_  |   3.18   |    2.85    |  **2.78**  |      2.83     |
> | Wine             |  _0.93_  |   0.97   |      1     |    0.97    |    **0.91**   |
> | Protein          |   2.87   |   3.12   |    2.93    |  **2.77**  |     _2.79_    |
> | Yacht            |   1.25   |  _0.73_  |    2.01    |    1.64    |    **0.66**   |
> | Avg. Rank        |   2.89   |   3.56   |      4     |   _2.44_   |     **2**     |

---

> > ### Comment · Reviewer_zA8F · 2022-08-03
> > **Thanks for author's response**
> >
> > Thank authors for the response and I appreciate the ablation experiment and added tabular data result. I maintain my score of 7 and vote for acceptance.

---

> > > ### Author Response · Authors · 2022-08-09
> > > **Thank You!**
> > >
> > > Thank you for going over the rebuttal and voting for the acceptance of this work!

---

### Official Review · Reviewer_KbkE · 2022-07-12

**Rating:** 6
**Confidence:** 3
**Soundness:** 2 fair
**Presentation:** 2 fair
**Contribution:** 3 good

**Summary:**

This paper proposes $\Delta$-UQ, a novel uncertainty estimation technique with a single neural network. The main idea is to ensemble predictions computed from inputs transformed by random anchors; to justify the use of ensembling by random anchoring, the authors analyze the Neural Tangent Kernel (NTK) of the predictions evaluated at inputs shifted by anchors and show that the NTK indeed changes according to the different anchors. Then this observation is further elaborated to introduce a single neural network model, $\Delta$-UQ, where the prediction is just computed from a single neural network but using different random anchors for each input. The proposed method is demonstrated to outperform existing uncertainty quantification methods for deep learning on various benchmark tasks.

**Questions:**

- Probably the most important part of the paper is the argument where the use of a single neural network on lifted space (the joint space $(c, x-c)$); but at least for me, it is unclear how this is justified. As far as I understand, what the paper is showing is as follows:
1) Derive the NTK for the shifted inputs (eq (2)) and linearized the infinite width limit of the prediction (eq (5)) in the original space ($(x-c)$).
2) Derive the same NTK and predictions on the joint space $(c, x-c)$ and show that those are similar to eq (2) and eq (5).
So what I get from this is that the NTK and predictions computed with $(x-c)$ and $(c, x-c)$ for a "specific" $c$ have similar forms; how this leads to the justification of "using a single $f$ but with multiple $(c, x-c)$ can simulate ensembling of multiple models trained with different $c$s? I'm pretty sure I'm missing something here.

- At least from the NTK argument there seems to be no restriction on the choice of $c$s, but the actual implementation of $\Delta$-UQ relies on setting $c$s as shuffled training set; wouldn't the algorithm work if $c$s are drawn from arbitrary distributions such as uniform or Gaussian distributions?

- If I understood correctly if we set $c$s to be shuffled batches during training, for the inference, we should reload the training set and draw $c$s from it to compute predictions (eq (6)). This can actually be quite impractical especially when the size of the training set is large (e.g., ImageNet). Can you comment on this?

- The main argument (NTK) assumes infinitely wide neural networks; does the proposed method work well for narrow neural networks?

- Does the proposed method work seamlessly with batch norm?

**Limitations:**

- The NTK argument is driven by a very restricted setting; equation (1) is only for the 2-layer MLP. I think that the same argument would not extend to the NTKs for arbitrary feedforward neural networks.
- Similarly, the linearization argument in equation (5) does not work for the loss function other than $L_2$ loss. The most apparent example would be classification models trained with cross-entropies, which seem to be commonly used settings for the experiments.
- There still is a gap between what the theory is assuming and the actual settings with which the neural networks are trained; e.g., NTK initialization and small learning rate. At least for me, the NTK argument does not seem very convincing to justify the empirical success of $\Delta$-UQ.
- While the experimental results on OOD detection tasks or corrupted data classification are quite convincing, there is no result on the standard image classification tasks for in-distribution data missing; how does $\Delta$-UQ perform on image classification benchmarks such as CIFAR and ImageNet in terms of accuracy, negative log-likelihoods, Brier scores, or expected calibration errors? Does it still outperform deep ensembles?

P. S.
There is an important relevant work worth mentioning - Teye et al., Bayesian uncertainty estimation for batch normalized deep networks, ICML 2018. Here, the authors propose a single neural network uncertainty quantification method where the epistemic uncertainty is derived by randomness in shift and scaling vectors for batch norm, and the prediction is done similarly to this paper where they randomly draw mini-batches from the training set, compute batch statistics, and ensemble the predictions computed from multiple batch statistics.

**Strengths And Weaknesses:**

Strength
- To the best of my knowledge, the idea of ensembling from multiple shifts (anchors) is novel, and it is also good to see that the proposed method works well with a single neural network.
- The experimental results are promising and convincing.

Weakness
- The main argument based on NTK does not correspond to the actual settings on which the neural networks are trained.
- It is rather unclear how the single neural network argument is justified (see below questions).
- No experimental results on the most standard task - image classification & uncertainty quantification for in distribution data.

---

> ### Author Response · Authors · 2022-08-01
> **Responses to Reviewer KbkE**
>
> We thank the reviewer for the detailed comment and useful suggestions to improve the paper.
>
> **Ensembling like behavior in the delta-UQ case.. NTK perturbation with different c’s**
>
> We showed the NTK perturbation for a single c in the paper for easier exposition -- however, this expression takes a similar form even in the case with different c’s,:
> $$
> [c_1,x_1-c_1]^{\top}[c_2,x_2-c_2] = x_1^{\top}x_2 + 2c_1^{\top}c_2 - c_1^{\top}x_2 - c_2^{\top}x_1
> $$
> $$
>  = x_1^{\top}x_2 - c_1^{\top}(c_1c_2^{\top} x_1 + x_2 - 2c_2)
> $$
> where, we exploit the fact that $c_1, c_2$ are normalized to be on the hypersphere. We can see that by setting $v  =  c_1c_2^{\top} x_1 + x_2 - 2c_2$, we can get a similar form of perturbation as $x_1^{\top}x_2 - c_1^{\top} v$ by combining all $c_1$ related terms as before  (and equivalently for $c_2$).
>
> Consequently, the perturbation analysis resembles what we have derived in the paper. Our hypothesis is that the simple coordinate transformation adopted by delta-UQ enables the use of multiple anchors (hence the stochasticity) against the same sample ‘x’. From the perspective of NTK, this has the effect of perturbing the NTK itself, inducing an ensemble-like behavior.
>
> **The choice of c’s is arbitrary, and it should work if c’s are drawn from arbitrary distributions such as uniform or Gaussian**
>
> The reviewer’s observation is correct -- the anchor c can indeed be drawn from arbitrary distributions (e.g., uniform or Gaussian) and as long as the “anchor distributions” are consistent across training & testing, our results should hold. As a simple design choice, we always choose the anchor distribution to be drawn from the batch itself at random. However, from our initial experiments, we observed that even random distributions are a valid choice.
>
> **Shuffled batches at training, during inference we need to reload train set -- impractical for large training datasets like ImageNet**
>
> Our inference only requires a very small number of anchors (10-20 anchors) in almost all cases. Hence, for inference, we can pre-select a random subset of anchors from the training data and do not have to draw random anchors for every single test example -- so from a memory overhead standpoint this is not significant. However, we agree with the reviewer that this requirement of storing the anchors could be a minor bottleneck in practice -- in such scenarios, using anchors drawn from a random distribution (like Gaussian or Uniform) can be an alternative solution.
>
> **The main argument (NTK) assumes infinitely wide neural networks; does the proposed method work well for narrow neural networks?**
>
> We use the NTK argument to analytically understand how anchor perturbations affect the NTK to induce an ensemble-like behavior. In practice, even the motivating example shown in Fig 3 of the main paper is performed with a simple MLP with just 10s of neurons. Even with such a narrow network, we notice that it is still able to produce meaningful epistemic uncertainty estimates. To demonstrate its broad applicability across network architectures with varying depths and widths -- we used MLPs (sequential optimization), ResNet-20 (CIFAR-10) and ResNet-50 (Imagenet) -- in our experiments. Finally, as part of our future work, we are planning to evaluate our insights (from infinite-width NTK analysis) empirically with finite-width NTKs (e.g., Fast Finite Width Neural Tangent Kernel, https://arxiv.org/pdf/2206.08720.pdf)
>
> **NTK argument is very restricted..would not extend to arbitrary NNs.. linearization argument only works for L2 loss not cross entropy**
>
> We use the NTK framework only to justify and study how anchor perturbations in fact induce an ensembling like behavior, which is a new observation of our work. We also use this to show how $\Delta$-UQ is an effective implementation of this stochastic data centering. As we have shown empirically with deep networks, our method is able to elicit accurate uncertainties with different loss functions. However, we agree that there is definitely more room for more rigorous theoretical/empirical analysis of the proposed approach and is definitely an important future direction.
>
> **Does the proposed method work seamlessly with batch norm?**
>
> Yes, all our image classification networks are unchanged (ResNet-50) etc. except for the initial layer to accept 6 input channels. These contain batch normalization that we do not modify -- and it does not affect the performance of our method.
>
> -- response continued in the next comment.

---

> > ### Author Response · Authors · 2022-08-01
> > **Part2: Response to Reviewer KbkE**
> >
> > **No result on standard image classification or in distribution” “how does $\Delta-$UQ perform on image classification like ImageNet and CiFAR?.. Does it still outperform deep ensembles?**
> >
> > Thanks for pointing this out. $\Delta-$UQ performs comparably to deep ensembles on in-distribution prediction performance and in terms of prediction fidelity under distribution shifts -- we have added new tables in the supplement for ImageNet, CIFAR-10, and regression performance on a suite of UCI benchmarks.
> >
> > 1. Results for NLL, Brier Score and ECE for CIFAR-10 in distribution and all shifted variants are added as a new figure in the supplement (Table 4, Figure  2). We see that $\Delta$-UQ performs better than Deep Ensembles nearly in all OOD settings, and comparably with in-distribution validation data. Here is the summary table for reference.
> >
> > **CIFAR-10 → CIFAR-10 validation, CIFAR-10C (varying corruption severity)**
> >
> > | **Method**  | **val**  | **1**    | **2**    | **3**    | **4**    | **5**    | **Avg**   |
> > |-------------|----------|----------|----------|----------|----------|----------|-----------|
> > | vanilla     |   90.5   |   81.8   |   75.1   |   68.3   |   60.6   |   49.1   |    69.8   |
> > | MCD         |    91    |   83.7   |   77.5   |   70.1   |   61.5   |   49.4   |    70.2   |
> > | SVI         |   88.6   |   82.3   |   76.9   |   70.8   |   63.1   |   52.6   |    70.6   |
> > | DEns        | **93.4** | **85.9** |  _79.8_  |  _73.1_  |   _65_   |  _52.4_  |   _72.9_  |
> > | $\Delta$-UQ |  _92.3_ | **85.8** | **80.9** | **75.2** | **67.8** | **56.2** | **74.25** |
> >
> > 2. For Imagenet, we followed the protocol from [1,2] (Table 3 in the supplement). Note, we could not obtain the NLL, Brier or ECE scores for in-distribution data from the official repository, other than in the summary plots provided in their papers. We have contacted the authors, and will include the results in a potential final camera ready version of the paper.
> >
> > **ImageNet → ImageNet Val, ImageNet-C (varying corruption severity)**
> >
> > | **Method**  | **val**  | **1**  | **2**  | **3**  | **4**  | **5**    | **Avg**   |
> > |-------------|----------|--------|--------|--------|--------|----------|-----------|
> > | vanilla     |  _76.1_  |  62.5  |   52   |   42   |   30   |   19.5   |    47.8   |
> > | MCD         |    75    |   60   |   50   |   38   |   29   |    17    |     46    |
> > | SVI         |  _76.1_  |  _63_  |   53   |   43   |   31   |    20    |   48.05   |
> > | DEns        | **78.1** | **66** | **56** | **47** | **36** |  **22**  | **50.05** |
> > | $\Delta$-UQ |  _76.1_  |  61.7  | _53.1_ | _44.2_ | _33.2_ | **21.8** |  _48.95_ |
> >
> > 3.. For UCI benchmarks, we follow the protocol from https://github.com/google/uncertainty-baselines/tree/main/baselines/uci and report the NLL metric for all cases (Table 2 in the Supplement). Here is the table for reference.
> >
> > **UCI Benchmark Regression Performance Evaluation**
> >
> > |  **UCI Dataset** |  **MCD** | **DEns** | **BNN-VI** | **PBP-MV** | **$\Delta-$UQ** |
> > |:----------------:|:--------:|:--------:|:----------:|:----------:|:-------------:|
> > | Boston           |  **2.4** |   6.11   |    3.12    |   _2.54_   |      2.58     |
> > | Concrete         | **2.93** |    3.2   |    3.22    |   _3.04_   |      3.09     |
> > | Energy           |   1.21   |  _0.61_  |    0.93    |    1.01    |    **0.56**   |
> > | Kin8nm           |   -1.14  |   -1.17  |    -1.03   |  **-1.28** |    _-1.19_    |
> > | Naval Propulsion |   -4.45  |   -5.17  |  **-6.12** |    -4.85   |    _-5.86_    |
> > | Power Plant      |   _2.8_  |   3.18   |    2.85    |  **2.78**  |      2.83     |
> > | Wine             |  _0.93_  |   0.97   |      1     |    0.97    |    **0.91**   |
> > | Protein          |   2.87   |   3.12   |    2.93    |  **2.77**  |     _2.79_    |
> > | Yacht            |   1.25   |  _0.73_  |    2.01    |    1.64    |    **0.66**   |
> > | Avg. Rank        |   2.89   |   3.56   |      4     |   _2.44_   |     **2**     |
> >
> > In case our answers have justifiably addressed your concerns, we respectfully request that you increase your score to support the acceptance of our work.
> >
> > Refs:
> >
> > [1] Ranganath Krishnan and Omesh Tickoo. Improving model calibration with accuracy versus uncertainty optimization. In NeurIPS 2020.
> >
> > [2] Yaniv Ovadia, Emily Fertig, Jie Ren, Zachary Nado, David Sculley, Sebastian Nowozin, Joshua Dillon, Balaji Lakshminarayanan, and Jasper Snoek. Can you trust your model’s uncertainty? Evaluating predictive uncertainty under dataset shift. NeurIPS 2019.

---

> > > ### Comment · Reviewer_KbkE · 2022-08-07
> > > **Thanks for the response**
> > >
> > > I appreciate the authors' effort to clarify my concerns; most of them are resolved, so I raise my score to 6.

---

> > > > ### Author Response · Authors · 2022-08-09
> > > > **Thank You!**
> > > >
> > > > Thank you for going over the rebuttal and voting for the acceptance of this work!

---

### Author Response · Authors · 2022-08-01
**Summary Response**

We thank the Area Chair and all the reviewers for providing high-quality feedback for our paper. We are glad that reviewers recognize our novelty and empirical benefits of $\Delta-$UQ. We hope our responses address all the concerns raised.

Summary of changes:
1. New results (CIFAR-10, Imagenet, UCI -- Table 2, Table 3, Table 4, Figure 2 in the updated supplement) demonstrating the accuracies of $\Delta-$UQ models in comparison to other existing strategies.
2. Additional ablation (Table 5) illustrating the importance of anchoring during inference
3. Expanded section 7 in the supplement on the GAN optimization experiment.
4. Illustration of the behavior of the uncertainties from $\Delta-UQ$ as training size increases.
5. Fixed a typo in Table 2 of the main paper (AUC score for BNN on Hartmann 3). Also, added the average rank to illustrate the overall performance of different UQ estimators.

In addition, we respond to all clarifications requested by the reviewers individually.

---

### Meta-Review · Area_Chair_gsbG · 2022-08-29

**Recommendation:** Accept
**Confidence:** Certain

**Metareview:**

The paper proposes a method that allows single model uncertainty estimation by training a model with a random data augmentation. The proposed approach is simple and scalable. It is comparable to or better than deep ensemble in terms of NLL, ECE, and Brier score. The application to sequential optimization tasks presented in the paper looks interesting. All reviewers support accepting this paper. While there could be more theoretical support, I think this paper would be of wide interest to the NeurIPS community. if possible, accept as Spotlight.

**Award:**

No

---

### Decision · Program_Chairs · 2022-09-14

Accept